# Predictive model for cytoneme guidance in Hedgehog signaling based on Ihog- Glypi-cans interaction

Adrián Aguirre-Tamaral [1,2], Manuel Cambón [2], David Poyato [2,3], Juan Soler [2] ✉ & Isabel Guerrero [1] ✉

During embryonic development, cell-cell communication is crucial to coordinate cell behavior, especially in the generation of differentiation patterns via morphogen gradients. Morphogens are signaling molecules secreted by a source of cells that elicit concentration-dependent responses in target cells. For several morphogens, cell-cell contact via filopodia-like-structures (cytonemes) has been proposed as a mechanism for their gradient formation. Despite of the advances on cytoneme signaling, little is known about how cytonemes navigate through the extracellular matrix and how they orient to find their target. For the Hedgehog (Hh) signaling pathway in *Drosophila*, Hh co-receptor and adhesion protein Interference hedgehog (Ihog) and the glypicans Dally and Dally-like-protein (Dlp) interact affecting the cytoneme behavior. Here, we describe that differences in the cytoneme stabilization and orientation depend on the relative levels of Ihog and glypicans, suggesting a mechanism for cytoneme guidance. Furthermore, we have developed a mathematical model to study and corroborate this cytoneme guiding mechanism.

Cell-cell communication is crucial during organism development, and it is mainly mediated through specific signaling molecules called morphogens. These signals are distributed in a graded form within a morphogenetic field, activating different target genes in a concentration-dependent manner[1]. Classical models assume that the graded distribution of signals occurs through simple diffusion[2], although for many morphogens their biochemical properties inherently impede their free diffusion through the extracellular matrix (ECM)[3]. For those cases, experimental evidence suggests a mechanism for morphogen transport based on its distribution by specialized signaling filopodia called cytonemes[4–7].

Cytonemes play a critical role in the process of cell communication in several biological systems, both during development and in tissue homeostasis in adults[8,9]. Thus, understanding the underlying mechanisms of cytoneme behavior turns essential. Here, we focus on the study of Hedgehog (Hh) signaling cytonemes in the *Drosophila* wing imaginal disc. This epithelium is divided into two compartments:[10] the posterior (P) compartment cells produce the Hh signal that is received by the anterior (A) compartment cells. In this system, cytonemes protrude from both producing and receiving cells, driving both short- and long-distance cell signaling[11,12]. The interaction of cytonemes from both compartments at specific contact sites, where all the components of the reception complex interact, promotes Hh reception[11].

Previous studies have identified several proteins involved simultaneously in Hh signaling and cytoneme dynamics[11]. A key protein is the Hh co-receptor Interference hedgehog (Ihog), an adhesion protein containing four immunoglobulin (IG) domains, two fibronectins type

[1]Tissue and Organ Homeostasis, Centro de Biología Molecular "Severo Ochoa" (CSIC-UAM), Nicolás Cabrera 1, Universidad Autónoma de Madrid, Canto-blanco, E-28049 Madrid, Spain. [2]Departamento de Matemática Aplicada and Research Unit "Modeling Nature" (MNat), Facultad de Ciencias, Universidad de Granada, E-18071 Granada, Spain. [3]Institut Camille Jordan (ICJ), UMR 5208 CNRS & Université Claude Bernard Lyon 1, F-69100 Villeurbanne, France. ✉e-mail: jsoler@ugr.es; iguerrero@cbm.csic.es

III (FNIII) domains, a single-pass transmembrane domain and an intracellular domain[13,14]. The overexpression of Ihog is able to stabilize (freeze) cytoneme dynamics. This stabilization has been described and quantified in[11,12,15], where cytonemes seem to be static.

Other proteins required for long-range Hh gradient formation and also present in cytonemes are the ECM components Heparan sulphate proteoglycans (HSPGs)[16,17]. In *Drosophila*, two glypicans belonging to the HSPGs family have been identified: Division abnormally delayed (Dally) and Dally-like protein (Dlp). Dally and Dlp also intervene in cytoneme dynamics and their interaction with Ihog facilitates cytoneme mediated cell contact[11]. The interaction of Ihog with Dally and Dlp occurs through the first fibronectin Fn1 domain of Ihog[15,18]. Thus, the Fn1 domain is involved in both Ihog-Hh and Ihog-glypicans heterophilic interactions, Ihog-Ihog homophilic interaction for cell-cell adhesion, and in the regulation of cytoneme dynamics. The double function of Ihog and glypicans in controlling both signaling reception and cytoneme behavior suggests that coordination between Hh reception, cytoneme behavior and their spatial orientation is essential for efficient signaling.

The dynamics of Hh cytonemes over time have already been studied[11,15,19] together with the correlation of dynamic cytonemes with the morphogen signaling gradient[12,20]. Here, we complement previous studies exploring how cytonemes are oriented and guided for the correct Hh signaling. In this work we observe that the interplay between different levels of Ihog and glypicans can orient and guide cytonemes in the Hh producing and receiving cells. To better understand this interplay, we develop a theoretical model that combines experimental data with mathematical tools and physical principles. Our model is able to predict cytoneme trajectories under different levels of Ihog and glypicans; these predictions have been experimentally validated in *Drosophila* wing imaginal discs. We propose that the glypicans-Ihog interaction can provide the spatial information required for cytoneme guidance during Hh reception.

## Results

### Ihog overexpression reveals cytoneme spatial behavior

Both the thinness and highly dynamic behavior of cytonemes hamper their observation in fixed tissues[12,21], so it is important to mention that the non-visualization of cytonemes in fixed tissues in not equivalent to the absence of cytonemes. Previous in vivo experiments in abdominal histoblast nest, where the visualization of temporal dynamic cytonemes is feasible in physiological conditions[12], have shown that the stabilization of cytonemes by overexpression of Ihog, in the same cells (overexpression in *cis*), is around 70% of the total cytoneme population (Supplementary Fig. 1a[11]) without changing their length and orientation (Supplementary Fig. 1b[11]). By stabilization we mean that cytonemes remain static ("frozen") for up to 9 hours[12], while their typical wild type lifetime is about 10 min[11]. We observed the same stabilization effect in ex vivo experiment in the wing imaginal disc (Supplementary Movie 1). This change in the dynamics of cytonemes, making them static, has been previously referred to as cytoneme stabilization by Ihog overexpression[11,12,15], probably forming bundles of two or more cytonemes[18]. This condition has been used as a genetic tool to visualize and study cytonemes in previous works[11,12,18,22] and we have used it here to study their spatial behavior.

In the wing imaginal disc, Ihog overexpression shows that cytonemes from both Hh producing and receiving cells protrude perpendicularly to the A/P compartment border (Fig. 1a[12]). This alignment is also present in wild type[12] (Supplementary Fig. 2), probably serving to facilitate the encounter of cytonemes protruding from the P compartment cells, carrying the Hh signal, with cytonemes protruding from the A compartment cells receiving it[11].

To experimentally dissect the orientation of cytonemes, we first confronted the P compartment cell population, overexpressing Ihog, to A compartment cell populations expressing different levels of Ihog.

We will refer as *trans* interaction to the effect on the behavior of cytonemes protruding from a cell population when it is confronted to a genetically different cell population. Cytonemes protruding from P compartment cells overexpressing Ihog are stabilized when confronted to A compartment cells in which Ihog levels are wild-type (Fig. 1a[12]) or downregulated (Fig. 1b[11]), while no cytonemes are observed when both confronted A and P cell populations equally overexpress Ihog (Fig. 1c). We quantified and compared the stabilization effect and orientation of cytonemes under the above conditions (Fig. 1d–f). To ascertain that the apparent lack of cytonemes in the confronted condition is due to non-stabilization instead of absence, we performed in vivo experiments in abdominal histoblast nets, and shown in Supplementary Movie 2, dynamic cytonemes are visible.

To quantitatively explore the spatial effect of Ihog on cytoneme behavior, we induced clones of different sizes and positions using a genetic tool that allows the random generation of two different types of confronted clones. In these clones, fluorescent labeling, as well as changes in Ihog levels, can be independently manipulated using the LexA and Gal4 drivers (Fig. 2a). These experiments showed that proximity is important for cytoneme stabilization (Fig. 2b). Cytoneme length is affected by the proximity of another clone if it is close enough (<15 microns Fig. 2c). This effect becomes statistically stronger the shorter the distance (Fig. 2c); actually, cytonemes in close proximity to a clone can only be stabilized if they change their orientation to circumvent the confronted clone (Fig. 2d).

We have also studied the dependence of cytoneme behavior on Ihog levels by confronting cells overexpressing Ihog in the whole P compartment with groups of cells expressing different levels of Ihog (ectopic cell clones) within the A compartment. If an innocuous label such as the actin reporter LifeActGFP[23] is used, cytonemes stabilized by Ihog overexpression in the P compartment are seen (Supplementary Fig. 3a), while they are not seen when the confronted clones express high levels of Ihog (Supplementary Fig. 3b yellow arrows). Furthermore, when Ihog overexpressing cells are confronted with clones ectopically expressing Ihog RNAi, thereby decreasing the levels of Ihog in *trans*, cytonemes are visualized oriented towards the A receiving cells (Supplementary Fig. 3c red and yellow arrows). These results indicate that differences in Ihog levels between cytoneme membranes in *trans* play a decisive role in cytoneme dynamics.

Based on these results, it could be proposed that the wild-type distribution of Ihog in the wing disc, strongly downregulated in the Hh receiving cells[24] (Supplementary Movie 3), could suffice to initiate cytoneme orientation from P to A cells; that is, from regions with high levels of Ihog in the P compartment towards regions with lower levels in the Hh receiving area of the A compartment (see Supplementary Movie 3). This hypothesis, solely based on the wild-type levels of Ihog, cannot be valid for those cytonemes that orient from Hh receiving cells towards Hh producing P cells, with higher levels of Ihog. Reasonably, other factors might also be involved in achieving cytoneme orientation. Although Hh could have been a good candidate, at the moment we lack convincing evidences of a possible role of Hh in cytoneme generation or dynamics[11].

### Glypicans influence in *trans* the behavior of cytonemes expressing ectopic Ihog

Cytoneme stabilization also depends on the heterophilic interaction between Ihog and glypicans[11,15,18]; such stabilization in wing disc cells requires the presence in *trans* (neighboring cells) of the HSPGs[12] and, more specifically, of glypicans[11]. To test whether glypicans might also be required within the same cells (interaction in *cis*), we induced *tout velue (ttv)* and *brother of ttv (botv)* double mutant clones while expressing also Ihog in the P compartment (Fig. 3a). *ttv* and *botv* are genes coding for enzymes that synthesize the Heparan Sulfate chains of the glypicans Dally and Dlp, so the effect of their absence on Hh signalling is equivalent to that of the lack of glypicans[16,17]. Cytonemes

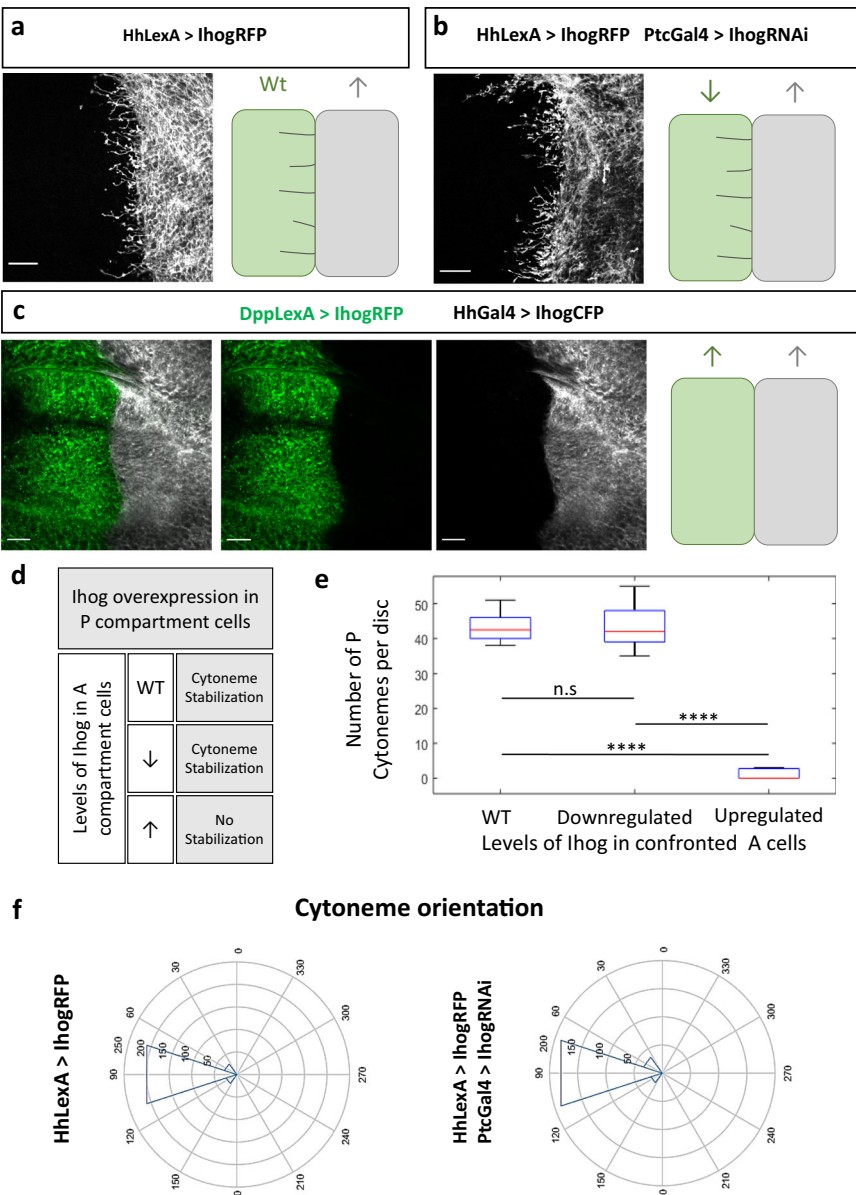

**Fig. 1 | Cytoneme stabilization by Ihog overexpression. a** P compartment cells overexpressing (↑) Ihog in *cis* (grey) are able to stabilize cytonemes over wild type (WT) A compartment cells (in *trans*). **b** P compartment cells overexpressing (↑) Ihog (grey) in *cis* are able to stabilize cytonemes over the A compartment cells (in *trans*) where Ihog is downregulated (↓). **c** P compartment cells overexpressing (↑) Ihog (grey) do not stabilize over the A compartment cells where Ihog is also overexpressed (↑, green) in *trans*. The schemes at the right depict the cytoneme behavior in the different genetic conditions. **d** Summary table depicting how cytoneme stabilization depends on Ihog levels in *trans*. e Boxplot and statistical study was performed using a Wilcoxon rank sum test per pairs over the number of P cytonemes per disc in different experimental conditions. **f** Orientation of P cytonemes in different conditions shows that P cytonemes are perpendicular to A/P compartment border. Data was quantified from $n = 23$ discs for section E and from 513 cytonemes of $n = 16$ discs for section F. In box plots, a box indicates the median (in red and 25 and 75 percentiles, whiskers indicate range of data and crosses indicate outliers. Raw data and p-values (ns:$p > 0.05$; *$p \leq 0.05$; **$p \leq 0.01$; ***$p \leq 0.001$; ****$p \leq 0.0001$) are provided as a Source data file Scale bars: 15 μm.

overexpressing Ihog emanating from HSPG-deficient cells (clone 2 in Fig. 3b, yellow arrowhead) are stabilized when navigating through wild-type territory, in contrast to cytonemes emanating from wild-type cells that are not stabilized when traversing glypican-deficient cells (clone 1 in Fig. 3b, red arrowhead)[12]. These data demonstrate that glypicans have an effect on Ihog-induced stabilization of cytonemes in *trans* but not in *cis*, most probably by enhancing the amount of Ihog in those cytonemes that go across glypican-deficient clones.

Accordingly, cytonemes are stabilized, probably in bundles, and oriented when a cell population with high levels of Ihog is in the proximity of another cell population (*trans*) with high levels of either Dally (Fig. 3c) or Dlp (Fig. 3d). However, cytonemes are not stabilized in

wild type regions if cytonemes come from a region co-expressing Ihog and Dally or Ihog and Dlp in the same cells (*cis*) (Fig. 3c', d'). These results suggest that cytoneme stabilization may be due to molecular competition dependent on the availability of Ihog and glypicans.

In this context, we observed that stabilization is recovered when a co-expressing region (Ihog + Dally or Dlp) is confronted to a region where either Dally or Dlp are overexpressed (Supplementary Fig. 4, red arrows). Interestingly, in those conditions we can observe how cytonemes are oriented to reach regions with high levels of Ihog and glypicans (Supplementary Fig. 4, yellows arrows). Therefore, there appears to be an orientation effect mediated by the Ihog-glypicans interaction in *trans* providing directionality to cytonemes.

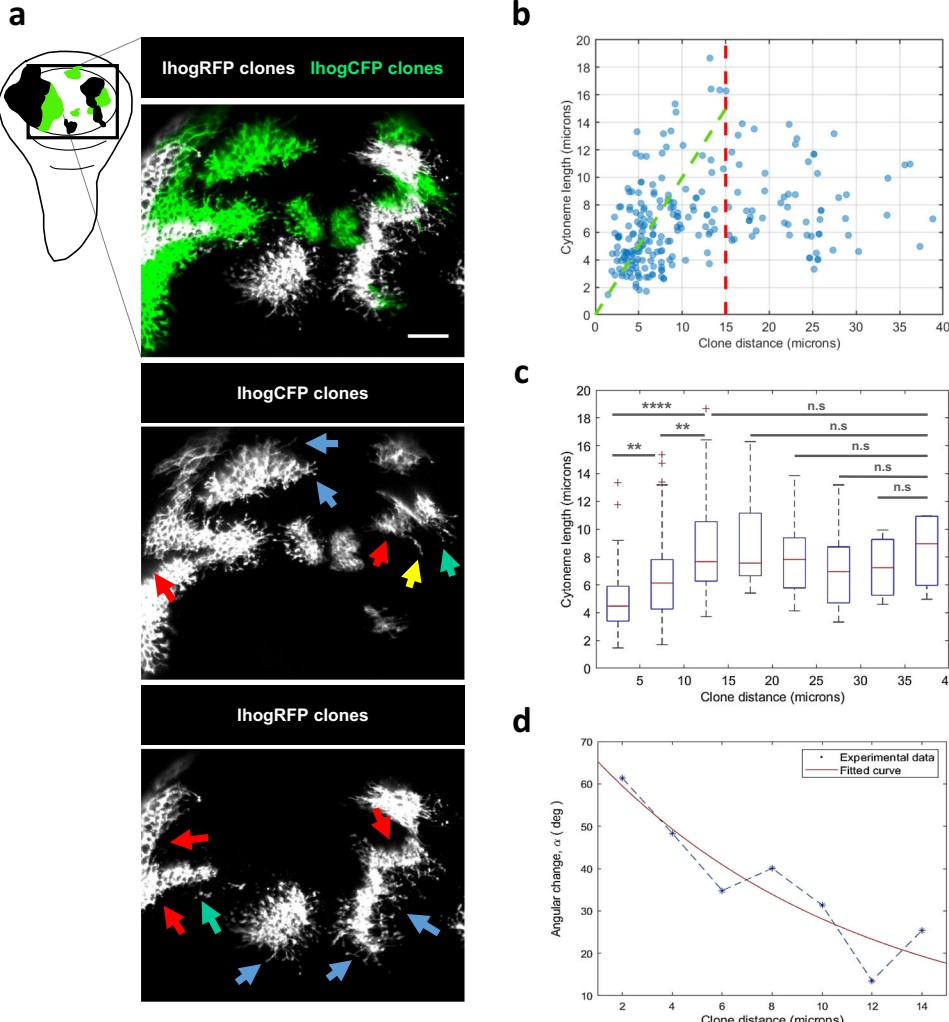

**Fig. 2 | Cytoneme behavior depends on the distance to the Ihog source. a** Ihog overexpression clones marked with different fluorophores. Blue arrows point to cytonemes stabilized towards areas without clones; red arrows point to touching clones showing no cytoneme stabilization when confronted; green arrows point to stabilized cytonemes crossing small clones (<2 cell diameters); <18% of the touching clones presented cytoneme stabilization (yellow arrows) for large clones (>2 cell diameters). **b** Cytoneme length (y-axis) versus distance between clones (x-axis) in μm. The red dotted line indicates the distances where the data cloud has a different behavior (<15 μm); the green dotted line represents the linear relation between the length of stabilized cytoneme and the distance between clones at short distances. **c** Statistical studies was performed using a Wilcoxon rank sum test

per pairs over different regions (boxplots of 5 μm) showed a significant change of cytoneme behavior if the clones were closer than 15 μm. **d** We studied in detail cytonemes over the green dotted line (i.e., stabilized cytonemes which are longer than the distance with the closest clone) and we observed that those cytonemes able to stabilize show a change in their orientation ($\alpha$) to avoid the confronted clones ($\alpha = 0$ means no change in orientation and $\alpha = 90$ a change towards opposite direction). Data were quantified along 235 cytonemes from different clones ($n = 21$). In box plots, a box indicates the median (in red) and 25 and 75 percentiles, whiskers indicate range of data and crosses indicate outliers. Raw data and *p*-values (ns:*p* > 0.05; *$p \leq 0.05$; **$p \leq 0.01$; ***$p \leq 0.001$; ****$p \leq 0.0001$) are provided as a Source data file. Scale bars: 15 μm.

## The Ihog Fn1 domain is essential for the *trans* interactions with Ihog and glypicans

Previous works have demonstrated that the cell adhesion of both the Ihog-Ihog homophilic and the Ihog-glypicans heterophilic interactions occur through the first Ihog fibronectin domain (Fn1)[15,18,22], the same domain involved in heparin- dependent Hh binding[22]. Therefore, to analyze the Ihog interactions leading to cytoneme orientation, we generated an Ihog form (LexAop.ihogΔFn1-RFP) deficient for the Fn1 domain, fused to RFP and under the control of the Lex inducible promoter. Supporting our previous results, we detected that the stability of cytoneme bundles observed in the *trans* Ihog-glypicans interaction (Fig. 4a[11]) is lost when the IhogΔFn1 is expressed (Fig. 4b). Furthermore, in contrast to the absence of Ihog cytoneme stabilization, observed after confronting two cell populations both overexpressing wild type Ihog (Fig. 1c), cytonemes can now be stabilized when confronting a cell population overexpressing wild type Ihog to a

cell population overexpressing the Ihog Fn1 deficient form (Fig. 4c). These changes in cytoneme behavior are statistically significant (Fig. 4d, e). Altogether, we conclude that the Ihog-glypicans heterophilic interaction might compete with Ihog-Ihog homophilic interaction for cytoneme stabilization and that both interactions require an intact Ihog Fn1 domain.

## Modeling cytoneme guidance: a mathematical framework

To further understand the mechanisms that control cytoneme orientation, we used a mathematical approach aimed at designing an in silico model able to predict cytoneme "guidance" based on the protein levels of Ihog, Dally and Dlp.

The experimental results presented above showed that the effects of those protein levels on the cytoneme spatial behavior could give rise to regions where cytoneme stabilization is favored compared to regions where it is not. These favored/disfavored regions are in some

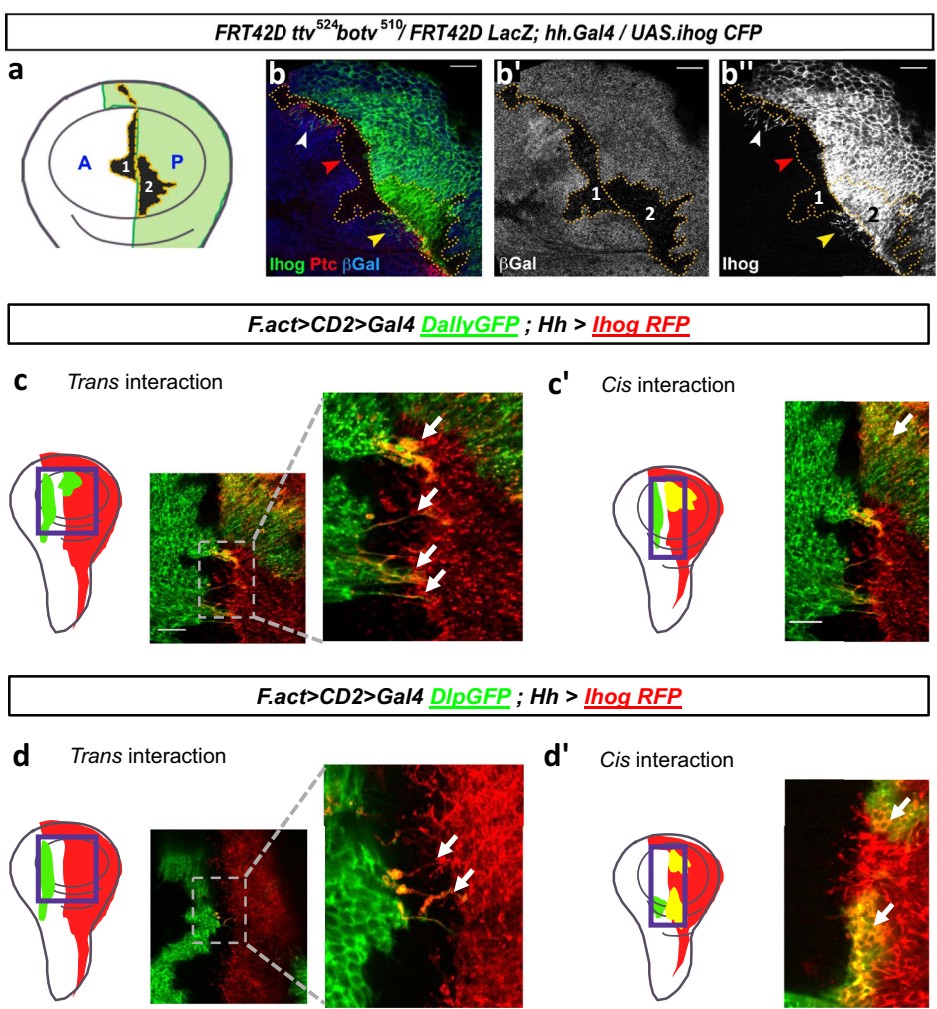

**Fig. 3 | Cytoneme behavior depends on Ihog and glycans levels. a** Scheme depicting induced *ttv⁻/⁻ botv⁻/⁻* double mutant clones in a wing disc while expressing also Ihog in the P compartment (green). **b** Wing disc containing induced *ttv⁻/⁻ botv⁻/⁻* double mutant clones marked by the absence of βgal (**b'**) while expressing Ihog in the P compartment (**b"**). Cytonemes stabilization can be observed (white arrow) in A compartment cells except in glycans loss of function clones induced in the A compartment cells, which do not express Ihog (red arrows). The yellow arrow indicates the presence of stabilized cytonemes when the *ttv⁻/⁻ botv⁻/⁻* double mutant clone is originated in the P compartment, which overexpresses Ihog, indicating that glycans are required in *trans* and not in *cis* for cytoneme stabilization. **c** Ihog and

Dally ectopic clones. Strong cytoneme stabilization is observed when Dally clones are confronted in *trans* interaction with P compartment cells overexpressing Ihog. **c'** Cytonemes are not visualized if Dally clones are in P compartment cells where Ihog is overexpressed (*cis* interaction). **d** Ihog and Dlp ectopic clones. **d** Strong cytoneme stabilization is observed when Dlp clones are confronted in *trans* interaction with P compartment cells overexpressing Ihog. **d'** Cytonemes are not visualized when Dlp clones are in the P compartment cells in which Ihog is overexpressed (*cis* interaction). Experimental sample size: $n_{Dally} = 17$ and $n_{Dlp} = 16$ discs. Scale bars: 15 μm.

sense reminiscent of the classical attractive/repulsive field in physics. With this idea in mind, we used a mathematical framework inspired by the physical theory of electric potentials where the "charges" in physics become "guiding factors" in our biological model. Thus, in our model, the optimal trajectories described by cytonemes across the tissue would be determined by the field lines associated to the "orientation" vector field, in which the levels of Ihog and glycans are the guiding factors generating the potential of the orientation field.

In what follows, we will briefly describe the building up of the orientation field; further details can be found in Methods and in Supplementary Notes. First, we outline the "charge" distribution associated to the guiding proteins, which serves as the source of the orientation potential that guides cytonemes. These sources will have different behaviors depending on the various levels of Ihog and glycans described for the wing disc pattern. The spatial dependence of protein distributions in the wing tissue emphasizes the importance of working with the complete 2D information. Accordingly, we define the "charge" distribution $\rho(\boldsymbol{r}, t)$ at the position $\boldsymbol{r} \in \mathbb{R}^2$ of the basal plane of the

imaginal wing disc epithelium as the total concentration of Ihog and glycans Dlp and Dally in tissue cell and cytoneme membranes, that is,

$$
\begin{aligned}
\rho(\boldsymbol{r},t) = {}& \beta_{\mathrm{Ihog}} \Big( [\mathrm{Ihog}](\boldsymbol{r}) + \sum\nolimits_{\mathrm{cyt}} [\mathrm{Ihog}]_{\mathrm{cyt}}(\boldsymbol{r},t)\delta_{\mathrm{cyt}}(\boldsymbol{r},t) \Big) \\
& + \beta_{\mathrm{Dlp}} \Big( [\mathrm{Dlp}](\boldsymbol{r}) + \sum\nolimits_{\mathrm{cyt}} [\mathrm{Dlp}]_{\mathrm{cyt}}(\boldsymbol{r},t)\delta_{\mathrm{cyt}}(\boldsymbol{r},t) \Big) \\
& + \beta_{\mathrm{Dally}} \Big( [\mathrm{Dally}](\boldsymbol{r}) + \sum\nolimits_{\mathrm{cyt}} [\mathrm{Dally}]_{\mathrm{cyt}}(\boldsymbol{r},t)\delta_{\mathrm{cyt}}(\boldsymbol{r},t) \Big).
\end{aligned}
\tag{1}
$$

where $[X](\boldsymbol{r})$ is the density of protein $X$ at the position $\boldsymbol{r}$ of the tissue cell membranes, $[X]_{\mathrm{cyt}}(r,t)$ is the density of protein $X$ at the position $\boldsymbol{r}$ of the cytoneme membranes, that can change spatially and temporally, and $\delta_{\mathrm{cyt}}(\boldsymbol{r},t)$ is the Dirac function that localizes the density along each cytoneme. "$\sum_{cyt}$" refers to the sum over all cytonemes protruding from A or P compartment cells. Note that, in particular, $\rho(\boldsymbol{r}, t)$ contains all the collective information of the guiding factors along all cytonemes. $\beta_{\mathrm{Ihog}}, \beta_{\mathrm{Dlp}}, \beta_{\mathrm{Dally}}$ are proportionality constants that govern the correct guiding factor attractive/repulsive relation among the implicated

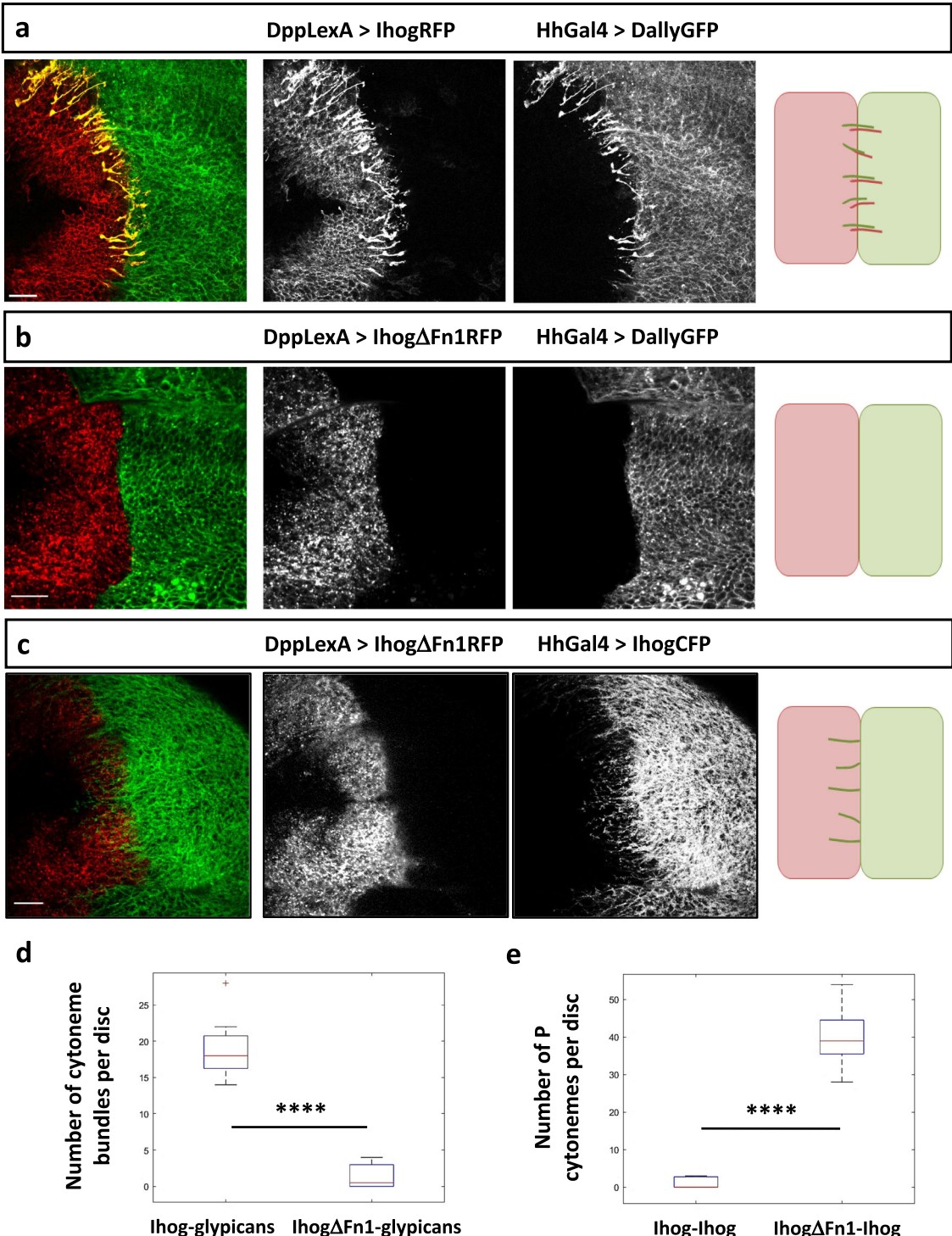

**Fig. 4 | Ihog and glypicans interaction for cytoneme stabilization in *trans* is mediated by the Fn1 domain of Ihog. a** Cytonemes protruding from both A and P compartments are strongly stabilized when cells overexpressing glypican Dally (green) are confronted to cells overexpressing wild-type Ihog (red). **b** Cytoneme bundle stabilization is lost when cells overexpressing Dally (green) are confronted to cells overexpressing the Ihog ΔFn1 mutant form (red). The schemes at the right depict the cytoneme behavior in the different genetic conditions. **c** Cytonemes protruding from cells overexpressing wild-type Ihog (green) are stabilized when confronted to cells overexpressing the Ihog ΔFn1 mutant form (red). **d** Boxplot and

the statistical study were performed using a Wilcoxon rank sum test per pairs over the number of cytoneme bundles per disc in the different experimental conditions. **e**) Boxplot and statistical study of the number of P cytonemes per disc in the different experimental conditions shown in Fig. 1c and panel c. Data were quantified from $n = 19$ discs for section D and from $n = 25$ discs for section E. In box plots, a box indicates the median (in red) and 25 and 75 percentiles, whiskers indicate range of data and crosses indicate outliers. Raw data and *p*-values (ns:$p > 0.05$; *$p \leq 0.05$; **$p \leq 0.01$; ***$p \leq 0.001$; ****$p \leq 0.0001$) are provided as a Source data file. Scale bars: 15 μm.

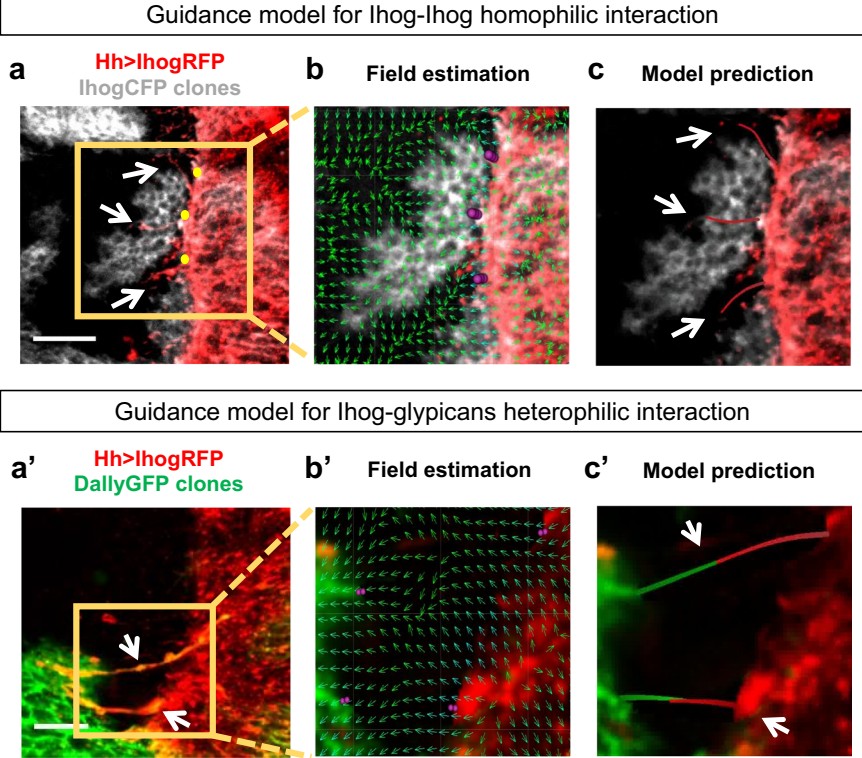

**Fig. 5 | Experimental validation of the guidance model. a** P compartment cells overexpressing Ihog confronted to clones also overexpressing Ihog. White arrows indicate those cytonemes to be simulated. **b** Guidance field estimation (green small arrows) of the experimental conditions eliminating selected cytonemes. **c** In silico predicted trajectories for the simulated cytonemes. **a'** P compartment cells overexpressing Ihog confronted with clones overexpressing Dally. White arrows show the cytonemes to be simulated. **b'** Guidance field estimation (green small arrows) of the experimental conditions eliminating selected cytonemes. **c'** In silico predicted trajectories for the simulated cytonemes. Scale bars: 15μm.

proteins (see the table of parameters in section 4 of the Supplementary Note).

The potential $\phi(\mathbf{r}, t)$, that contains the collective information for the orientation of cytonemes, is generated by the protein distribution $\rho(\mathbf{r}, t)$ as follows:

$$\phi(\mathbf{r},t)=(K^*\rho)(\mathbf{r},t)=\int K(|\mathbf{r}-\mathbf{s}|)\rho(t,\mathbf{s})\,d\mathbf{s}. \quad (2)$$

Here, the potential kernel function $K$ defines the type of interactions between cytonemes according to the guiding factors and must be experimentally determined. In our case, since cytonemes only sense protein levels in the nearest cell membranes (Fig. 2b, c), the potential kernel function $K$ must be of short-range. This means that the guidance information is spatially limited to the neighboring cells (Fig. 2c). Even though this short-range phenomenon does not take place in the classical electric field, it is in agreement with most of the cases described in biological systems[25].

The choice of the sign in parameters $\beta_{\text{Ihog}} > 0$, $\beta_{\text{Dlp}} = \beta_{\text{Dally}} < 0$ in the charge distribution (1), and orientation field (3) has been carefully set in agreement with the above experimental results, that is, the observed effects of the Ihog-glypican interactions and their consequences on the stabilization and orientation of cytonemes. We refer to the discussion in section 1 of the Supplementary Note for further details.

Finally, from the potential $\phi(\mathbf{r},t)$ we can obtain the orientation vector field $\mathbf{O}(\mathbf{r},t)$ that guides cytonemes:

$$\mathbf{O}(\mathbf{r},t)= -\nabla\phi(\mathbf{r},t)= -\int \nabla K(|\mathbf{r}-\mathbf{s}|)\rho(t,\mathbf{s})d\mathbf{s}. \quad (3)$$

The dynamics of cytonemes is then deduced from a principle of least action. In other words, the equation of motion of each cytoneme arises as a minimizer of the Lagrangian associated to its kinetic and the above (2) potential energy under the natural constrains of the biological system. This leads to a coupled system of non-linear and non-local hyperbolic equations for each cytoneme, which allows modeling the guidance process (see equation 12 in the Supplementary Note).

## In silico study of cytoneme orientation under overexpression conditions of Ihog and glypicans

In order to validate our model, we predicted in silico the trajectories of cytonemes protruding from experimental clones overexpressing Ihog (Fig. 5a) or glypicans (Fig. 5a') induced in the A compartment when confronted to P compartment cells expressing Ihog.

For simulations, we selected some of the experimental cytonemes (arrows in Fig. 5a–a'), removed them and predicted their orientation according to our model and within the simulated field (Fig. 5b–b'). Note that the orientation field corresponds to the frozen configuration after cytoneme removal and that the new simulated cytonemes grow and evolve in response to the global Ihog concentration.

The model simulations were able to predict the correct cytoneme orientation under conditions of overexpression of Ihog and glypicans (Fig. 5c–c'). It is important to note that the trajectories of simulated cytonemes are nearly identical to the experimental ones, even the curvature and orientation tendency were similar in both the in silico simulations (Fig. 5c–c') and in the real cases (Fig. 5a–a'). The temporal evolution of the cytonemes can be observed in more detail in Supplementary Movies 4 and 5.

## Model prediction of guidance for wild type cytonemes

Since our guidance model is able to predict cytoneme orientation in Ihog, Dally and Dlp overexpression conditions, we explored its prediction capacity for cytoneme orientation in wild type wing discs.

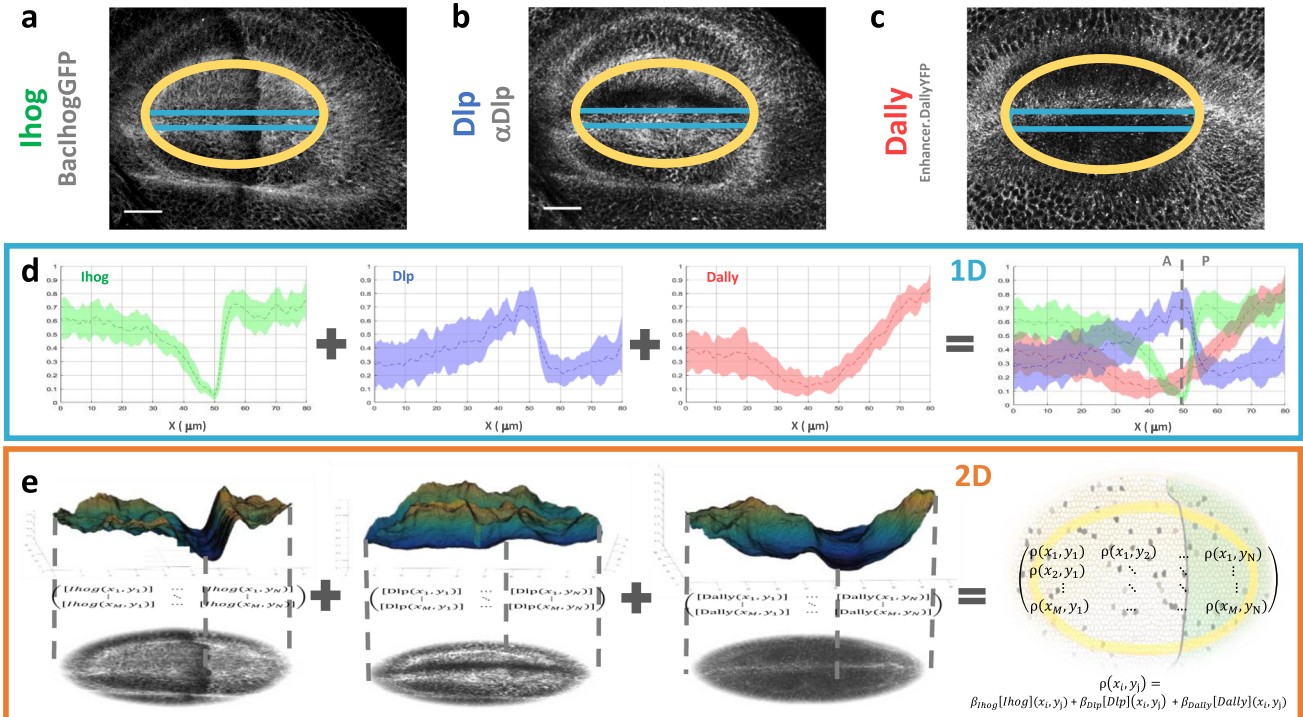

**Fig. 6 | Wild type Ihog, Dally and Dlp spatial distributions at the basal plane of wing imaginal discs. a** Wild type Ihog basal distribution in a Bac.IhogGFP wing disc. **b** Wild type Dlp basal distribution obtained by immunohistochemistry. **c** Wild type Dally basal distribution in a Dally.trap.YFP wing disc. **d** Characterization of Ihog (green), Dlp (blue) and Dally (red) profiles at the basal plane of imaginal wing discs. Dotted lines represent the average profile and shaded area and their experimental variability, quantified by the standard deviation over different samples ($n_{Ihog}$ = 17, $n_{Dlp}$ = 17, $n_{Dally}$ = 19). **e** Surface representation of wild-type Ihog, Dlp and Dally basal patterns in the marked elliptical area of the wing pouch. These quantified 2D patterns (matrices) are the numerical inputs for the computational wild-type simulations. Scale bars: 30 μm.

To this end, the wild type spatial distribution of each protein was analyzed by fluorescence in Bac.IhogGFP wing discs (Fig. 6a), using the antibody for Dlp (Fig. 6b) and in Dally.trap.YFP (Fig. 6c) wing discs. Representative samples of their protein distributions have already been characterized in 1D profiles[24]. Here we have made a statistical study of the profiles in several samples and examined the spatial correlation between Ihog, Dally and Dlp (Fig. 6d, 1D profiles). We observed that Ihog has a strong downregulation in the Hh reception region, Dlp increases its levels on the same region, and Dally is downregulated at both sides of the A/P compartment border.

A quantitative description in terms of 2D spatial distribution is needed for more accurate model simulations. To that end, we quantified the 2D spatial distribution of the basal protein pattern from confocal fluorescence images. In particular, we developed a script that allows visualization of the 2D protein patterns of the wing imaginal disc on 3D surfaces (Figs. 6e, 2D profiles) and generation of numerical data (matrices in Fig. 6e) for the levels of each protein, which have been used in (1) to model cytoneme orientation in wild type conditions. The 2D quantification showed that the wild-type modulation of the protein levels is not the same in all regions (Figs. 6e, 2D profiles), something that could not be deduced from 1D profiles.

We then tested if, using our guidance model, the wild-type spatial protein distributions of these proteins could be sufficient to predict cytoneme orientation within the Hh signaling region (Fig. 7). Our simulations show an increase in the parallel orientation of the field lines in the A compartment cells close to the A/P border (Fig. 7b, small red arrows) that change direction and become perpendicular to the A/P compartment border. Furthermore, as depicted in Fig. 7d, the model is able to predict the real observed cytoneme orientation in wild type cells (Fig. 7b).

To further study cytoneme behavior at the A/P compartment border region, we selected six cells in the Hh signaling area, three from the A and three from the P compartment. Cytonemes have a dynamic behavior of extension and retraction[11], although the signals triggering these dynamics have not yet been identified. For the simulations, we have considered two different plausible behaviors of retraction: cytonemes retracting after having contacted the tip (middle bottom cytonemes in Supplementary Movie 6) and cytonemes that retract after contacting other cytoneme while growing (middle top cytonemes in Supplementary Movie 6). The resulting predictions (Fig. 7d and Supplementary Movie 6) show that cytonemes protruding from the Hh-producing cells of the P compartment (green) are oriented towards the Hh receiving cells of the A compartment (red) to deliver the morphogen. At the same time, cytonemes protruding from receiving cells are oriented towards the producing cells to collect the morphogen.

## Discussion

Hh graded distribution across the receiving *Drosophila* epithelia is mediated by cytonemes[12]. The transport of Hh is composed of two parallel and complementary processes. On the one hand, the transport along the cytonemes of vesicles containing Hh is associated with other proteins[26,27]; on the other, the elongation, orientation and contact of these cytonemes for reception[11,12]. In this work, we have proposed a model focused on the orientation of cytonemes.

It has been described that receptors, coreceptors and ligands are present in close proximity for the contact-dependent Hh reception involving both source and recipient cell cytonemes, in a way similar to a synaptic process[11,27,28]. The purpose of contact would be the release of Hh at specific cytoneme sites. The molecular and cellular events that induce the interactions between cytonemes are critical determinants

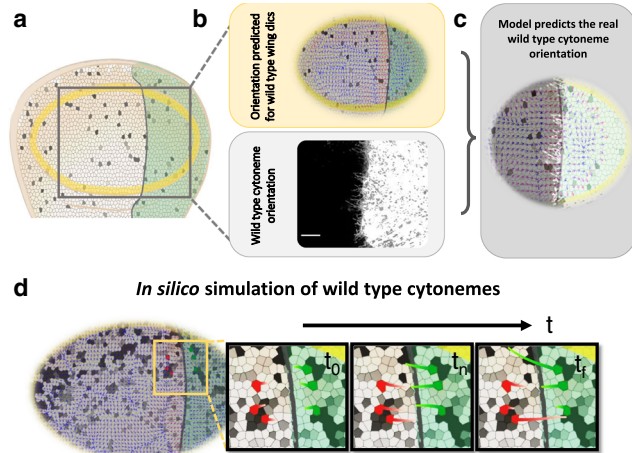

**a** Scheme depicting a wing imaginal disc. **b** Guidance field prediction (small arrows) in a wing disc using experimental wild-type distributions of Ihog, Dally and Dlp (top). Wild type orientation of P compartment cytonemes, marked with CD4-tomato (bottom). **c** The model predicts cytoneme orientation in wild-type conditions. **d** In silico predictions for cytoneme temporal growth on wild-type discs.

**Fig. 7 | In silico simulation of cytoneme orientation in wild-type wing discs.**

of signaling. Therefore, a cytoneme orientation and guidance mechanism must be needed for directed and efficient signaling. In this context, we have described that, in the Hh reception area, cytonemes are oriented perpendicular to the A/P compartment border. This preferential orientation, clearly shown by overexpressing Ihog, would give the shortest and most efficient contact-dependent reception. Using a multidisciplinary approach, we have developed a biomechanical guidance model to explain cytoneme orientation during Hh signaling.

The adhesion molecule Ihog plays a role in cytoneme dynamics since its overexpression stabilizes cytoneme temporal dynamics in different *Drosophila* tissues[11,15,19]. Here, we have analyzed if Ihog could also be involved in the control of cytoneme guidance. We first studied the effect of Ihog after confronting populations of cells expressing different levels of Ihog. The results have shown that cytoneme stabilization depends not only on the differential levels of Ihog in opposing cell populations but also on the distance between populations expressing different Ihog levels. We have observed that cytonemes are stabilized by Ihog overexpression even when Ihog is not present in the confronted cell population, indicating that Ihog-Ihog homophilic interaction in different cell populations is not sufficient for cytoneme orientation.

Since Dally and Dlp are also extracellular components of the Hh pathway, we investigated their possible involvement in the orientation of cytonemes. These proteins interact with Ihog for Hh signaling[21,24,29], cytoneme stabilization[11,15] and cell adhesion[18,22]. We studied cytoneme guidance using cytoneme stabilization as a tool and confronting clones with different levels of glypicans and Ihog. Cytonemes were stabilized and oriented whenever a cell population with high levels of Ihog was in the proximity of another cell population with high levels of either Dally or Dlp. However, cytonemes protruding from clones co-expressing Ihog and Dally or Ihog and Dlp were not stabilized. These results suggest a different cytoneme behavior depending on whether Ihog and glypicans interaction occurs in *trans* (in different cells) or in *cis* (in the same cells). Accordingly, glypican loss of function clones in the wing imaginal disc also indicates that *cis* interactions of Dally and Dlp with Ihog are not needed for Ihog-induced cytoneme stabilization, while *trans* interactions are necessary. Therefore, the specific protein levels of Ihog and glypicans in *cis* and *trans* can regulate both cytoneme stability and orientation. We then propose that the differences in levels of Ihog and glypicans could be responsible for cytoneme

orientation at the A/P compartment border and that these differences might be due to molecular competition of Ihog-Ihog homophilic and Ihog-glypicans heterophilic interactions. The "freezing" effect on cytonemes is probably due to Ihog sequestering glypicans in *trans*. In addition, the observed lack of cytoneme stabilization in confronted cell populations overexpressing Ihog might indicate that an excess of Ihog sequestering the glypicans available in *cis* could block the stabilization of the opposing cytonemes in *trans*.

It has recently been described that Ihog-Ihog interaction for cell-cell adhesion occurs via the fibronectin FNIII domains[18], which are also responsible for the Ihog-glypican interaction: after expression of an Ihog mutant form lacking the Fn1 domain (IhogΔFn1), the ability of Ihog to form homodimers and to interact with glypicans disappears[15,18]. We have observed here that: 1) There is no cytoneme stabilization after expressing IhogΔFn1, in contrast to the cytoneme stabilization seen in cells overexpressing wild type Ihog. 2) Cytoneme stabilization, absent when Ihog-overexpressing cell populations are confronted, is recovered when cytonemes from cells overexpressing wild type Ihog are confronted to cells expressing IhogΔFn1. 3) Due to the lack of interaction, no strong stabilization is observed when cells expressing IhogΔFn1 are confronted to cells overexpressing glypicans.

The Fn1 domain has been described to be essential for both Ihog-Ihog and Ihog-Hh interactions[18]. Since the purpose of cytoneme contact recognition at reception is the release of Hh at specific cytoneme contact sites, the competition between Ihog-Ihog homophilic interaction and Ihog-Hh heterophilic interaction has been proposed to regulate the reception process at contact sites[22]. Interestingly, the expression of an Ihog form with point mutations in the Fn1 domain that abrogate the interaction with Hh but not with glypicans stabilizes and orients cytonemes as the overexpression of wild-type Ihog does[15,18]. Therefore, although both Ihog-Hh and Ihog-Ptc interactions are essential for Hh transfer during reception, Hh by itself does not seem to be necessary for the generation or the stabilization of cytonemes[11]. However, the expression patterns of Ihog, Dlp and Dally are regulated in Hh signaling regions, probably due to a modulation by the Hh pathway itself[24,30]. Consequently, a regulatory loop might be established in which the Hh responses could indirectly influence the orientation of cytonemes. This autoregulatory loop is consistent with the one described for the orientation of cytonemes in the FGF signaling pathway in *Drosophila*[31–33].

Our experimental results permit the generation of a framework for cytoneme guidance dependent on different levels of Ihog, Dally and Dlp. To this end, using a multidisciplinary approach, we designed a mathematical model based on the potential theory in physics, which resembles the observed favored/disfavored regions for cytoneme stabilization. We validated our model comparing the in silico predictions with the experimental results obtained from clones expressing different levels of Ihog and glypicans. The model simulations were able to reproduce with high accuracy the experimental cytoneme trajectories.

To close the circle, after validating our model in gain of function conditions, we studied the orientation of cytonemes observed in the wild-type wing imaginal disc. Our model predictions showed that the wild type protein distributions of Ihog, Dally and Dlp can generate a field of directions that can be enough to orient both producing and receiving cytonemes. In other words, for proper cytoneme guidance, the interactions of glypicans and Ihog might allow the exchange of information about cellular relative locations. Therefore, we propose a new role of Ihog and glypicans in cell-cell signaling, influencing the orientation of cytonemes for an efficient contact during the reception process. Inefficient cytoneme contacts could explain the lack of Hh responses in ttv[−/−] btv[−/−] double mutant clones located in the reception region[16,17].

Our model is general enough to be applied to predict and study cytoneme guidance in other signaling processes. For instance, we have observed that cytonemes are also oriented perpendicular to the D/V

compartment border, where Wg and Notch, but not Hh, are expressed[34]. This could suggest that Wg and Notch signaling can be dependent on cytonemes in the wing disc, as it has been shown in other *Drosophila* tissues[35–37] and in vertebrates[38–40]. Since glypicans have a specific expression pattern at the D/V border[41], they might help the driving of cytonemes perpendicular to this border.

Likewise, our model can also be applicable to vertebrate systems. Other receptor proteins with structural domains similar to Ihog, such as DCC and Robo, have been described to be important in axon guidance[42], what emphasizes the similarities between cytoneme-mediated cell signaling and neuronal communication[11,27,28]. Moreover, the role in neuronal guiding of some extracellular matrix proteins, including glypicans, has been described[43,44]. Similar orientation of cytonemes after ectopic expression of CDO and BOC (vertebrate homologs of Ihog and Boi) has been reported for Shh signaling in the chicken limb bud[45], suggesting that the control mechanisms to orient cytonemes might be evolutionarily conserved.

## Methods

### Experimental material and methods

**Drosophila stock maintenance.** *Drosophila melanogaster* stocks were maintained according to protocols described in Ashburner manual[46]. In particular, flies were reared under standard lab conditions at 25 °C in vials and fed with yeast for their life circle. For transient expression of transgenic constructs, the fly crosses were maintained at 18 °C and then incubated at restrictive temperature (29 °C) to inactivate the Gal80ts repressor 24 hours before dissection. The description of mutations, insertions and transgenes is available at Fly Base (http://flybase.org).

**Binary expression systems.** The transient expression (overexpression or downregulation) of proteins was performed under the control of the binary systems: Gal4/UAS[47] LexA/LexAop[48]. Although each binary system works in a similar manner, each system only recognizes its specific target sequences. This specificity allows the combination of both systems in the same sample.

For those cases in which there are not specific drivers to express the transgenes in a specific tissue region, or in cases in which the mutant phenotype is very invasive, it is possible to study the mutant condition limited to certain cell populations (clones). This is achievable through the use of the FLP/FRT system[49], which uses the enzyme flippase (FLP) to induce recombination between FRTs (Flippase Recombination Target) sequences. The mitotic recombination was induced through a heat shock (HS) at 37⁰C at a specific developmental stage and the *Drosophila* tissue cells respond by generating at random mosaic recombinant clones with a genotype different than the rest of the tissue. In combination with the previous binary system, we generated random clones in which the transgenes are activated. In this work, we have used the CoinFLP system[50], which is a useful Flip-out recombination system that introduces another FRT (FRT3) to allow the induction of two types of clones, each one with a different driver. In our case, we have used CoinFLIP-LexA::GAD.Gal4 to induces clones expressing either the Gal4 or the LexA drivers.

The following drivers were used to induce ectopic expression using the Gal4/UAS and LexA/LexO systems: tubGal80ts, hs-Flp122 (Bloomington Drosophila Stock Center, BDSC), hs-FLP[49], hh.Gal4[51], F.Act > CD2 > Gal4[52], Ptc Gal4[53], ap.Gal4[54], hh.LexA[11], generated by Ernesto Sánchez-Herrero, CBMSO), CoinFLIP-LexA::GAD.Gal4 (BDSC 59270) and dpp.LexA[55].

### Overexpression stocks
**The UAS-transgene strains.** UAS.ihog-YFP[21], UAS.ihog-CFP[29], UAS.CD4-tdTomato[56], UAS.ihog-RNAi (VDRC 102602), UAS.boi-RNAi (VDRC 108265), UAS.LifeActGFP (BDSC 35544), UAS.LifeActRFP (BDSC 58362), UAS.dlp-GFP[57], UAS.dally-GFP[58], UAS.GMA-GFP (Actin-binding

domain of moesin tagged with GFP)[59] and UAS.CD63-Cherry (Provided by Clive Wilson, Oxford University).

**The LexAop-transgene strains.** LexAop.ihog-RFP[11]. The LexAop.ihogΔFn1-RFP construct used IhogΔFn1-RFP from the pTWR vector[15], which was introduced into the pLOTattB plasmid[48] carrying the lexA operator (LexAop). Transgenic strains were recovered using standard protocols.

**Other stocks.** Dally.trap.YFP (DGRC 115511), EnhancerPtcRed (Kyoto stock center, DGRC 109138), Bac.IhogGFP: Rp49 promoter[30], ttv [524][17]. and botv[510] [17]. Since the Ihog homolog, Brother of Ihog (Boi) can substitute Ihog in some functions[60], Ihog downregulation experiments were also done by expressing together Ihog (ref: 102602) and Boi (ref: 108265) RNAis (VDRC; http://stockcenter.vdrc.at).

**Transient expression of transgenes.** The binary system can be used under a temporal control of the GAL80ts that acts as a Gal4 repressor blocking the interaction between Gal4 and the UAS sequence. The GAL80ts have also been described to repress the LexA/LexAop system when it contains the Gal4 sequences repressible by GAL80. For transient expression of transgenic constructs, the crosses were maintained at 18 °C and then incubated at the restrictive temperature (29 °C) to inactivate the Gal80ts repressor 24–48 h before dissection. After dissection, third instar imaginal discs were fixed in 4% paraformaldehyde (PF) in PBS for 20 min at room temperature (RT) and mounted in Vectashield medium (Vector labs).

Mutant clones were generated, as described, by a flipase induced by heat shock (HS) incubating larvae at 37 °C. The HS was given at 48–72 h after egg laying (AEL) for 7–10 min in the case of the CoinFLIP-LexA::GAD.Gal4 system and for 10–15 min in the case of the F.Act > CD2 > Gal4 system. $ttv^{-/-} botv^{-/-}$ double mutant clones ($ttv^{524} botv^{510}$) were generated using FRT42D cassette under the control of hsFlp [122] flippase and individuals were incubated at 37 °C for 45 min.

**Immunohistochemistry of wing imaginal discs.** Immunostaining was performed according to standard protocols[61]. Imaginal discs from third instar larvae were fixed in 4% paraformaldehyde (PFA) in PBS for 20 min at room temperature (RT) and permeabilized with 0,1% Triton in PBS (PBT) before incubation with 1% BSA (bovine serum albumin) in PBT for blocking the tissue to decrease the unspecific background (1 h at RT) and follow by primary antibody incubations (overnight at 4 °C). Incubation with fluorescent secondary antibodies was performed for 1 h at RT and then washed and mounted in mounting media (Vectashield).

The mouse monoclonal α-Dlp primary antibody (from[13]) and the rat monoclonal α-Ci primary antibody (from[62]) were used at 1:30 and 1:20 dilutions, respectively, for regular immunostaining and/or extracellular immunostaining. The protocol for the extracellular labelling was done following[63]; imaginal discs from third instar larvae were dissected on ice, transferred immediately to ice-cold M3 medium containing α-GFP (rabbit anti GFP polyclonal antibody from Chromotek, ref: PABG1, at 1:1000 dilution) and α-Dlp (mouse monoclonal antibody from[13] at 1:30 dilution) antibodies and incubated at 4 °C for 1 hour. The incubation with the primary antibody was done under these ex vivo conditions, without detergents prior fixation, preventing antibody penetration inside cells. Imaginal discs were then washed in ice-cold PBS, fixed in PBS 4% PF at 4 °C, washed in PBT and incubated with the secondary fluorescent antibody as above. The secondary antibodies used in this work were anti-mouse Pacific Blue from ThermoFischer (ref: P-31582) at 1:400 dilution and anti-rabbit Alexa 647 from ThermoFischer (ref: A-31573) at 1:400 dilution.

**Data acquisition for fixed, in vivo and ex vivo experiments.** Laser scanning confocal microscopes (LSM700 and LSM800 Zeiss) were

used for confocal fluorescence imaging of the fixed imaginal discs. The in vivo abdominal histoblasts imaging was performed in a chamber to seat and orient the pupae to look under the microscopy as described in[64]. The dorsal abdominal segment A2 was filmed using 40x magnification; Z-stacks of around 10 μm of thickness with a step size of 0.8 μm were taken, every 2 min, using a LSM800 confocal microscope. For the ex vivo experiments we induced 40 h of Ihog overexpression in P posterior cells and we dissected third instar larvae as described in[64]. The ex vivo imaging was performed in standard microscope slides, where the discs were mounted in M3 insect medium and covered with a 15 × 15 mm coverslip. The samples were filmed using 40x magnification; Z-stacks of around 8 μm of thickness with a step size of 0.62 μm were taken every minute using a LSM800 confocal microscope.

**Data analysis and statistical study.** The data of abdominal histoblast nests are published in[11]. The quantification of the number of cytonemes and their orientation angles have been obtained manually using Fiji. The data representation, in boxplots and rose polar diagrams, was performed using Matlab2015a functions. For the statistical study of the data in the figures, we first performed a Shapiro–Wilk test of normality, since the S-W test showed that our data distributions were not parametric, to study the statistical significance between datasets, we used a Wilcoxon rank sum test per pairs in Matlab2015a (See Supplementary excel file for the quantified data and the calculated *p*-values). Finally, some cytonemes were marked in ex vivo and in vivo experiments to better show the dynamics and the stabilization of cytonemes, those tracks were performed using the Fiji plugging: Manual Tracking.

**Quantification of cytoneme orientation.** The experimental data of cytoneme length and orientation due to Ihog overexpression in LexA and Gal4 clones were quantified manually using FIJI along 235 cytonemes from different clones ($n = 21$). In each measurement, we quantified the length ($\lambda$) and the angle ($\beta$) of a cytoneme and the distance ($d$) and angle ($\gamma$) of the tip of that cytoneme with the closest clone. The study of the effect in the length was performed analyzing $\lambda$ vs $d$, and the study of the orientation was calculated using $\alpha = \gamma - \beta$. Please notice that the increase in $\alpha$ means that cytonemes are avoiding the closest confronted clone. Limit cases, $\alpha = 0$ and $\alpha = 90$, can be respectively interpreted as no change in the direction and opposite direction.

**Quantification of Ihog, Dally and Dlp protein distribution profiles.** The profiles of each protein were obtained measuring, with Fiji profile tool, the fluorescence intensity over a ROI of 15–80 μm$^2$ in $n = 17$ Bac.IhogGFP wing discs, where the Dlp profile was also measured in the same ROI using α-Dlp primary antibody[13] and Dally profiles were obtained from the same ROI size in $n = 19$ Dally.trap.YFP wing discs. (See supplementary excel file for the quantified raw data). The 1D average profiles, together with the experimental variability (measured with the standard deviation of the quantified profiles), were computed following the mathematical protocol described in[20].

**Theoretical material and methods**
Since cytonemes mainly move over the basal plane of the tissue we can simplify our model to two spatial dimensions as considered above. In addition, since Ihog and glypican proteins are attached to cell membranes, we assumed that the guiding factors are spatially constrained to cytonemes and cells. In order to deduce the cytoneme kinetics (behavior), we have chosen the variational principle approach, which facilitates a robust mathematical formulation of the equations. Note that cytonemes are dynamic structures and the amount of proteins located on those cytoneme membranes changes spatially over time. This implies that the orientation field is dynamic and not only the protein concentration at cell membranes is responsible for the orientation of cytonemes, but also the motion of cytonemes themselves can

modify the generated field. This feedback loop, coupled with the biological complexity of the process, introduces a large non-linearity in the model, which forces us to solve the equations via numerical methods. Here, we briefly describe the theoretical and numerical protocol used for in silico simulations, a detailed description can be found in Supplementary Note.

Note that the dynamics of cytonemes cannot be reduced simply to each single protein moving freely under the force field $\boldsymbol{O}(\boldsymbol{r}, t) = -\boldsymbol{\nabla}\phi(\boldsymbol{r}, t)$. In fact, we must take into account the geometric information that filopodia structure imposes spatial constraints to those proteins that are attached to their membranes. We return here to the aforementioned idea suggesting that energy must be optimized during the movement of cytonemes according to the principle of least action. Let $L(t)$ be the variable length of the curve $\boldsymbol{\gamma}(\xi, t)$ that represents the cytoneme. Then, the minimization problem

$$\min_{\gamma \in \mathscr{D}} \int_0^T \int_0^{L(t)} \left( \underbrace{\frac{1}{2}|\dot{\gamma}(\xi, t)|^2}_{kinetic\ energy} - \underbrace{\bar{\rho}\phi(\gamma(\xi, t), t)}_{potential\ energy} \right) d\xi\, dt,$$

$$\mathscr{D} = \{\gamma \in C^1 : |\gamma'(\xi, t)| = 1, + \text{boundary conditions}\},$$

leads to the differential equation that governs the dynamics of the cytoneme, (see supplementary note for a more exhaustive description):

$$\frac{\partial}{\partial t}(\dot{\boldsymbol{\gamma}}) = \frac{\partial}{\partial \xi}(\lambda \boldsymbol{\gamma}') - \frac{1}{\tau}\dot{\boldsymbol{\gamma}} - \bar{\rho}\boldsymbol{\nabla}\phi(\boldsymbol{\gamma}, t), \quad (4)$$

being $\tau$ the relaxation time due to linear friction of the cytonemes with the extracellular matrix. Here, $\bar{\rho}$ is the distribution of proteins over the specific cytoneme $\boldsymbol{\gamma}$, whilst the orientation potential $\phi$ is collectively generated by all protein distribution in the system according to (1).

In the sequel, we present a discretized version of the dynamics of cytonemes that allowed us to computationally solve the previous continuous model (4). In order to do this, we defined the discretized cytoneme as an inextensible chain of $N(t)$ connected nodes carrying a specific protein concentration. Then, we used generalized coordinates in order to eliminate inextensibility constraints. Hence, the position $\boldsymbol{r}_i(t)$ of the $i$-th node of a cytoneme $\boldsymbol{\gamma}(\xi, t)$ at time $t$ is determined from the position $\boldsymbol{r}_{i-1}(t)$ of the $(i - 1)$-th node by $\boldsymbol{r}_i(t) = \boldsymbol{r}_{i-1}(t) + l_i(\cos\theta_i(t), \sin\theta_i(t))$, for some angle $\theta_i(t)$ and some length $l_i$ between nodes $i$ and $i - 1$. Then, we obtain:

$$\boldsymbol{r}_i(t) = \boldsymbol{r_0} + \sum_{j=1}^{j=i} l_j(t)\left(\cos\theta_j(t), \sin\theta_j(t)\right), i = 1, \ldots, N(t), \quad (5)$$

where $\boldsymbol{r_0}$ denotes the base point where the cytoneme is attached to the cell membrane. Note that since cytonemes can elongate and retract, the length $l_{N(t)}$ of the last piece of the chain can grow (up to a maximal length) or decrease, and the number $N(t)$ of nodes will change with time. Then, we can discretize the continuous differential Eq. (4) at each discrete parameter value $\xi_i = \sum_{j \leq i} l_j$ representing the $i$-th bond $\boldsymbol{r}_i(t) \approx \boldsymbol{\gamma}(t, \xi_i(t))$. Then, using the change of variables in (5), and the generalized coordinates $\boldsymbol{\Theta}(t) = (\theta_1(t), \ldots, \theta_{N(t)}(t))$ and $\dot{\boldsymbol{\Theta}}(t) = (\dot{\theta}_1(t), \ldots, \dot{\theta}_{N(t)}(t))$ the constrained dynamics of the whole discrete cytoneme in generalized coordinates are:

$$\boldsymbol{M}(\boldsymbol{\Theta}(t), t)\ddot{\boldsymbol{\Theta}}(t)^T = -\frac{1}{\tau}\boldsymbol{M}(\boldsymbol{\Theta}(t), t)\dot{\boldsymbol{\Theta}}(t)^T + \boldsymbol{G}\left(\boldsymbol{\Theta}(t), \dot{\boldsymbol{\Theta}}(t), t\right)^T. \quad (6)$$

Here $\boldsymbol{M}$ and $\boldsymbol{G}$ are a matrix and a vector, respectively, whose explicit expression is derived in the Supplementary Note. In particular,

***G*** contains all the collective information of the orientation potential generated by the full network of cytonemes. An alternative but equivalent derivation of (6) could have been to implement a discrete version of the principle of least action. Specifically, for the discrete kinetic energy $E_{Kin}\left(\boldsymbol{\Theta}(t),\dot{\boldsymbol{\Theta}}(t)\right)$ and potential energy $E_{Pot}(\boldsymbol{\Theta}(t),t)$ in generalized coordinates, we can formulate the discrete minimization problem of the corresponding discrete action functional $\mathscr{A}=E_{Kin}-E_{Pot}$ and (6) arises as its corresponding Euler-Lagrange equation. Although in this paper, for simplicity, we have set the relaxation time $\tau$ as a constant, we could also introduce modulation factors that modify $\tau$. For example, it might include the required time to stabilize the cytonemes in the membrane transmitting the information to other cells. In that case, the appropriate modulation should depend, among other factors, on the concentration of Hedgehog that the cytoneme carries, the Ptc concentration of the cell membrane, the co-receptor Ihog or other proteins with adhesion properties.

The matrix Eq. (6) contains the dynamics of each cytoneme. In our numerical simulations, we solved it numerically by first inverting ***M*** with LU-decomposition and then computing the time evolution of the guidance with a Runge-Kutta 4th order method for all the cytoneme nodes at the same time (see Supplementary Movie 7 for an illustrative example of the numerical resolution).

**Experimental data and model inputs for computational simulations - Model simulations for clone experiments.** To study and validate our model prediction in mutant conditions, we first selected an experimental clone, obtained previously by confocal imaging. We then selected an area of interest in the clone, that contained the cytonemes that we wanted to simulate. In order to maintain the closest conditions to the experimental case, the selected cytonemes were removed from the image with FIJI tools with the following protocol:

- Select the cytoneme area manually with the Freehand ROI tool.
- Save the ROI.
- Save the cytoneme origin position.
- Measure the average and standard deviation of the signal in the surrounding background area around previous ROI.
- Multiply the cytoneme ROI by zero to remove the cytoneme concentration.
- Add the average background signal to the ROI.
- Introduce a Gaussian noise with the standard deviation background value.

The resulting image is the one obtained experimentally but devoid of cytonemes. Next, we introduced the protein concentration of the experimental clones in (1) using the pixel distribution, and we simulated the path that each removed cytoneme would follow according to our model, departing from positions of origin previously saved.

**- Model simulation for wild-type wing imaginal disc.** The experimental protein concentrations ρ (1) were obtained from the pixel intensities in each channel in gray-scale units of the experiments described in the Results section. For wild-type simulations, the protein spatial distribution was measured in the elliptical area at the basal plane of the wing pouch; the elliptical shapes were scaled to the same lattice to be uploaded in the in silico wing disc. Dlp pattern was quantified using Dlp monoclonal antibody in the wing imaginal disc expressing either Ihog tagged with GFP (Bac.IhogGFP) or Dally tagged with YFP (Dally.trap.YFP). Extracellular staining was performed for Dally tag and Dlp protein to confirm that the same pattern is present in the extracellular matrix where the cytonemes sense the levels of glypicans and Ihog.

Since our tissue model works in 2D and Ihog and glypicans proteins are present along all tissue membranes, we obtained the quantitative description of 2D spatial distribution at the basal plane of the imaginal wing disc, plane where cytonemes are present. To that end,

the data were treated using a computational script that allows extracting the experimental data from confocal images and visualizing it in a 3D surface the 2D spatial distribution of the particular fluorescence protein pattern along the disc tissue. The script also generates numerical matrices that contain the quantifications of the levels of a specific protein in different regions. Those matrices are the experimental inputs that we used in our model to simulate in silico cytoneme orientation.

### Reporting summary
Further information on research design is available in the Nature Research Reporting Summary linked to this article.

### Data availability
Source data generated in this study is provided with this paper in the Source Data file, and more supplementary information from the corresponding authors upon request. Source data are provided with this paper.

### Code availability
The cytoneme simulation code was performed using a custom-made code on C# language using Unity Engine. The source code is available at https://github.com/AdrianA-T/Guidance-Model. More details of this computational code can be obtained from the corresponding authors on request.

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

## Acknowledgements

We are grateful to and Pedro Ripoll and Ana-Citlali Gradilla for comments on the manuscript. We also thank to Laura González-Méndez, Carlos Jiménez-Jiménez, David Sánchez-Hernández and Eléanor Simon in I.G laboratory for their help with some experiments, the Confocal Facilities of the CBMSO and Bloomington and Vienna stock centers for fly stocks. This work was supported by grants BFU2017-83789-P, PID2020-114533GB-C21 and TENTACLES consortium RED2018-102411-T to I.G from the Spanish Ministry of Science, Innovation and Universities and by institutional grants from the Fundación Areces and Banco de Santander to the CBMSO. FPI fellowship from the Spanish Ministry of Science, Innovation and Universities supported A.A-T (BFU2017-83789-P). This work was also supported by grants RTI2018-098850-B-I00 to J.S from the MINECO-Feder (Spain), PY18-RT-2422 & A-FQM-311-UGR18 to M.C, D.P and J.S from the Junta de Andalucia (Spain), MECD (Spain) research grant FPI2015/074837 to M.C, and partially supported by the MECD (Spain) research grant FPU14/06304 and the European Research Council (Europe) Project ERC-COG-2019 WACONDY (grant agreement No 865711) to D.P.

## Author contributions

A.A.-T. performed *Drosophila* experiments, immunostaining and imaging of confocal images of Drosophila imaginal disc and in vivo abdominal histoblast nets, as well as data quantification and statistical analysis, and wrote the manuscript. M.C. and D.P. developed the mathematical model and wrote the supplementary note. M.C. developed the computational code and simulations. J.S. designed and supervised the mathematical framework and wrote the manuscript. I.G. designed and supervised the *Drosophila* experiments and wrote the manuscript.

## Competing interests

The authors declare no competing interests.
