## [Peer Review File · Nature Communications]

Predictive model for cytoneme guidance in Hedgehog signaling based on Ihog- Glypicans interactionREVIEWER COMMENTS

Reviewer #1 (Remarks to the Author):

This paper describes cytonemes mediated cell-cell communication by combining experimental and computational approaches. The authors focus on stabilization and orientation of cytonemes to further understand how cytonemes navigate through the extracellular space to find their targets. The authors previously found that Hedgehog co-receptor Ihog and glypicans, Dally and Dlp, stabilize and guide cytonemes. In this paper, the authors found that orientation and guidance of cytonemes are regulated by interplay between different levels of Ihog and glypicans in Hh producing and receiving cells. They also developed a mathematical model to predict cytoneme orientation under Ihog and glypican interactions. The computational model is based on experimental data and physical principles. A series of excellent studies by the authors' group (2017 eLife, 2021 eLife) advanced the understanding of dynamics and mechanisms of cytonemes. The paper is a continued study and provides some interesting data. However, I found a huge gap between experimental part and modeling. The authors may consider formatting the paper into two distinct stories.

Major points

- The authors repeatedly use "stabilized cytonemes" in the paper although all the data has been collected after fixation of the tissue. Upon the analysis of fixed tissue images, the authors may argue frequencies of cytoneme generation, length and guidance, however I do not agree that the data support stability.
- The authors obtained cytoneme orientation data in abdominal histoblast and wing imaginal disc (Fig. 1 and S. Fig. 1). These data indicate that cytoneme dynamics might be very different between abdominal histoblast and wing imaginal disc. Although it is not clear to me whether the authors obtained the parameters for modeling from the data in abdominal histoblast due to insufficient information, if so, the parameters are not appropriate for modeling in wing imaginal disc.
- The authors discussed about the orientation of cytonemes. According to the data in Fig. 2 and Fig. 3, orientation of cytonemes appears to be instructed by the location of Ihog expression, and further regulated by the glypican expression. While the cytoneme formation is observed around the clones in Fig. 2, how are the multi-directional cytoneme formation instructed? Does gradient of Ihog or glypicans instruct the direction? Or other components such as chemotactic factors are needed? These are the important issues prior to establishing a model.

Minor points

- S. Fig. 1A: what do "stabilized cytonemes" mean? In the previous paper (2017 eLife), dynamics of cytonemes have been measured. However, I don't find any description of how the authors measure "stability" in either 2017 paper or current manuscript.
- Lane 116: "no stabilized cytonemes are observed", if I understand correctly, the authors only observed fixed tissue image in the wing imaginal disc. The data can be interpreted that generation or stability of cytonemes become less, but time-lapse analysis is needed to conclude which is the case. Although the authors have observed cytoneme dynamics in abdominal histoblast, the dynamics of cytonemes in wing imaginal disc may not be the same. Similar arguments are found throughout the paper.
- S. Fig. 2B, C: orientation of cytonemes should be statistically analyzed.
- Fig. 2: Very difficult to understand how the authors analyze the data. According to the picture in Fig. 2A, cytoneme formation might be influenced by the position of the clones (in the center or peripheral of the wing pouch). This has to be discussed.

- Fig. 6: How is the data processed? There is no clear description in methods. Providing that cytonemes are localized at the basal side of the wing epithelial cells, collecting basal specific Ihog, Dally and Dlp could be useful.
- S. Fig. 1: Details of the data collection are missing. How many samples tested? How were they measured?
- Eq-4 (lane 564) was argued but I could not find Eq-4 in the text.
- Fig. 4 legends are wrong.

Reviewer #2 (Remarks to the Author):

This manuscript aims to study the underlying mechanisms of cytoneme behavior by focusing on the Hh signaling cytonemes in *Drosophila* wing image disc. They first showed some experimental evidence by manipulating the levels of Ihog and Glypicans in regions of wing disc cells that the levels of Ihog and glypicans directly affect the cytoneme stabilization, and Ihog and glypicans interact in trans through the Fn1 domain of Ihog to affect cytoneme orientation. Then they developed a mathematical model predicting the effect of Ihog-glypicans interaction in cytoneme orientation.

While the results showing the effect of the levels of Ihog and glypicans on cytoneme behavior is interesting, several critical points remain to be addressed, such as whether Ihog overexpression induced cytoneme formation or stabilized cytonemes? Do those Ihog-induced/stabilized cytonemes from A or P compartment represent Hh signaling cytonemes? According to the model proposed here, cytoneme guidance only depends on the levels of Ihog, Dally and Dlp, the question is how the patterns of Ihog, Dally, and Dlp expression levels are created? And how about other factors, such as receptors and ligands? What's their contribution to the cytoneme behavior?

A recent paper (Yang, Zhang, Yang, et al. eLife 2021;10:e65770) showed interesting results on Ihog mediating cytoneme stabilization through trans-homophilic binding, which is supported by glypicans and occurs through the Fn1 domain of Ihog. Another recent study (Du et al, 2021, BioRxiv) on cytoneme-mediated FGF signaling provides experimental evidence that cytoneme pathfinding is a result of polarized bidirectional signaling response via cytonemes arising from direct FGF-FGFR interactions at the contact sites between the source and recipient cells. Authors should give a better introduction on this and other related findings, and clearly discuss what's the novelty of their findings on top of the previous studies.

Altogether, this paper presented interesting results, predicting a model that might explain one of the ways by which cytonemes are guided for Hh signaling.

Below are some point-to-point comments in detail.

Figure 1 related:

1. Page 4 line 103-108: "In the imaginal wing disc, cytonemes from both Hh producing and receiving cells protrude perpendicularly to the A/P compartment border, as seen following Ihog cis-overexpression (Fig 1A, F and 20105). Thus, Ihog overexpression reveals the spatial cytoneme behavior during the reception process, when cytonemes protruding from the P compartment cells contact with cytonemes from the A compartment cells." --- Can authors clarify what's the cytoneme behavior in wing discs without Ihog overexpression? Do you see cytonemes orienting toward A/P compartment border in the wt wing disc in live imaging conditions? Ex vivo imaging of live cultured wing disc is feasible. Also, the authors need

to explain how the cytoneme behaviors in this study were measured? Are they all done in fixed tissues?

2. when both confronted A and P cell populations equally express Ihog, there were no cytonemes observed, this result is not equivalent to "no stabilization".

Figure 2 related:

1. Page 5 line 135-141: "These experiments showed that, besides differential Ihog levels between clones, proximity and size are also important for cytoneme stabilization (Fig. 2B). We again observed that cytonemes protruding from confronting clones both expressing high levels of Ihog could not be stabilized (Fig. 2A), unless the clones were smaller than two cell diameters (Fig. 2C). The length at which a cytoneme can become stable depends on the distance between clones with high levels of Ihog (Fig. 2B)." --- This part of data is not clearly shown by the presented figures. Authors need to present more convincing data for the statements. For example, authors need to show more evidence for each type of interaction, and how many samples and clones were checked? In addition, from the presented image in 2A, it seems that IhogCFP clones polarize cytonemes toward each other from distance, rather than projecting cytonemes in all directions to the region without Ihog expression. Lastly, did the authors see any difference in cytoneme behaviors from Ihog-clones in different compartments?

2. Page 5 line 142-146: "Following this line of thought, we could propose that the wild type distribution of Ihog in the wing disc, where it is strongly downregulated in the Hh receiving cells, might suffice to initiate cytoneme orientation from P to A compartment: from regions with high levels of Ihog in the P compartment towards regions with lower levels in the Hh receiving area of the A compartment", --- authors need to provide a reference for this.

Figure 3 related:

Page 6 line 169-173: "Further suggesting an in trans effect over membrane stability by Ihog overexpression, clones overexpressing glypicans in both P and A compartments showed that P cytonemes are stabilized and oriented when a cell population with high levels of Ihog is in the proximity of another cell population (trans) with high levels of either Dally (Fig. 3C) or Dlp (Fig. 3D)." --- This sentence should be rewritten.

Page 7 line 184-200, --- The result, the Ihog Fn1 domain is the site for Ihog-Ihog and Ihog-glypicans trans interactions, has been shown in earlier studies (Yang, Zhang, Yang, et al. eLife 2021;10:e65770). Also, the authors concluded that Ihog-glypicans heterophilic interaction competes with Ihog-Ihog homophilic interaction for cytoneme stabilization, it remains unclear which interaction dominates or has higher affinity. Authors should provide more supporting evidence for this. Lastly, according to Fig. 4A, the number of cytonemes under Ihog-glypicans trans interaction is less than the number of cytonemes in Ihog expression condition like in Fig. 1A, and it is also less than the number of cytonemes with IhogDeltaFn1RFP expression in one side. Why this is the case? can this be explained by the proposed model?

Reviewer #3 (Remarks to the Author):

The authors investigate oriented stabilization as a driving source for the observed directionality of cytonemes, long cellular threads that are required for Hedgehog (Hh) signaling in multiple contexts, including Hh morphogen signaling in the Drosophila wing imaginal disc. The paper is largely based on (and guided by) previous data showing that the co-receptor Ihog and the two Drosophila HSPGs of the

glypican family Dally and Dlp, are required for Hh signaling and stabilize cytonemes. The study adds on the spatial requirements for these factors and shows that differences in the relative levels of Ihog and Dally/Dlp are instructive for the stabilization and orientation of cytonemes. Specifically, they find that cytonemes are stabilized, and potentially oriented, when cells of the epithelium display differences in the Ihog levels and experimentally flattening such differences cause loss (or destabilization) of cytonemes. Glypicans, previously shown to interact with and affect surface stability of Ihog, do also affect cytoneme stability and orientation in trans. This effect is mediated by their interactions with Ihog, suggesting that the relative levels of the players are important for cytonemes stabilization and orientation. The authors back up their data with an elegant mathematical model (to my knowledge the first application of mathematical modeling in the cytonemes field) which supports the findings and makes the prediction, that the physiologically occurring relative levels distribution of the components (Ihog, Dally, Dlp) could suffice to generate the observed cytoneme orientation and alignment.

Major points

Many of the data shown confirm or expand already previous findings (in part from the same group). The in vivo studies are very elegant and the tools used superb. In all experiments, the eventual read-out is the presence (i.e. stabilization of cytonemes) using fluorescent Ihog as a proxy. Thus, in all cases, overexpressed Ihog-GFP (or RFP/CFP) serves a dual function: It stabilizes cytonemes so that they become (easily) detectable, and at the same time it is used as a read-out for these structures. In other words, we are not looking at "wild-type" cytonemes, but rather at cytonemes induced by non-physiologically high levels of Ihog. While I understand that cytonemes of wild-type cells (in this case cells that are not overexpressing factors that impact on cytoneme dynamics) are hard to observe, I believe it would be essential to close the circle and test in at least in one assay the predictions of the models by monitoring cytonemes emanating from cells that are not additionally modified in their capacity to produce or stabilize cytonemes. Similarly, I miss some direct connection to graded Hh signaling itself using markers of the pathway.

I am not sure if I understand the interpretation of the experiments with IhogDeltaF1: Is this construct biologically inert? Can this construct when used in I the simplest form of the assays used here (hh>IhogDeltaFn1 similar to 1A) stabilize cytonemes? If not, then the interpretation of 4C is trivial (or just confirmatory that the protein is not functional).

Other/Minor points

Line 29 abstract: "A paradigmatic example" sounds redundant (= "An exemplary example"), consider "A prominent/classic/iconic example or paradigm?"

Figure 1 B and F: Here they use ptcGal4 instead of the dpp driver used throughout the study. Is there a reason for that? I guess both drivers express in the same cells. Neither of the drivers appears in the material section

Line 167: "navigating within glypican-def clones" this is unclear to me, do they mean cytonemes stemming from glypican-deficient clones?"

Sentence starting at line 195: Fix syntax

Reviewer #1 (Remarks to the Author):

This paper describes cytonemes mediated cell-cell communication by combining experimental and computational approaches. The authors focus on stabilization and orientation of cytonemes to further understand how cytonemes navigate through the extracellular space to find their targets. The authors previously found that Hedgehog co-receptor Ihog and glypicans, Dally and Dlp, stabilize and guide cytonemes. In this paper, the authors found that orientation and guidance of cytonemes are regulated by interplay between different levels of Ihog and glypicans in Hh producing and receiving cells. They also developed a mathematical model to predict cytoneme orientation under Ihog and glypican interactions. The computational model is based on experimental data and physical principles. A series of excellent studies by the authors' group (2017 eLife, 2021 eLife) advanced the understanding of dynamics and mechanisms of cytonemes. The paper is a continued study and provides some interesting data.

However, I found a huge gap between experimental part and modeling. The authors may consider formatting the paper into two distinct stories.

Major points

- The authors repeatedly use “stabilized cytonemes” in the paper although all the data has been collected after fixation of the tissue. Upon the analysis of fixed tissue images, the authors may argue frequencies of cytoneme generation, length and guidance, however I do not agree that the data support stability.

Previous studies have shown that changes in cytoneme stability caused by Ihog overexpression “freeze” cytonemes, making them easily detectable (González-Méndez et al., 2017; Bischoff et al., 2013). These experimental conditions have been used to analyze cytoneme behavior because, even after tissue fixation, those dynamical structures are very difficult to visualize unless the dynamics were slowed down by ectopic expression of Ihog or by other experimental tools. In the new version of the manuscript, we have included an *ex vivo* experiment confirming the stabilization of cytoneme (new Supplementary movie 1).

We agree with the Referee that guidance and cytoneme length are parameters that can be directly measured in fixed tissue. Indeed, the correlation of cytoneme length between Ihog overexpressing clones is one of the measures included in the new Figure 2. Following the reviewer's comment, we have also increased the quantification of guidance considering the angles of orientation (new Figure 2).

- The authors obtained cytoneme orientation data in abdominal histoblast and wing imaginal disc (Fig. 1 and S. Fig. 1). These data indicate that cytoneme dynamics might be very different between abdominal histoblast and wing imaginal disc. Although it is not clear to me whether the authors obtained the parameters for modeling from the data in abdominal histoblast due to insufficient information, if so, the parameters are not appropriate for modeling in wing imaginal disc.

The measurements of cytoneme dynamics in abdominal histoblast nests were obtained from published works (González-Méndez et al., 2017; Bischoff et al., 2013). Here, we only performed statistical studies based on these works.

Hh signaling is comparable in abdominal histoblast nets and in wing imaginal discs; both systems present similar anterior and posterior compartmental distribution (Hh producing and receiving cells) and comparable cytoneme-mediated Hh gradient formation (Bischoff et al., 2013; P.A. Lawrence et al., 1999). The new *ex vivo* experiment shows the equivalence of both systems, while clarifying the critics on cytoneme stabilization in wing discs. We agree with the Reviewer that the temporal scale in each system could be different, but dynamics is not the objective of this study.

For the *in silico* model, quantification of the parameters was performed in fixed wing imaginal discs. Following the reviewer's comments with respect to the model inputs and the simulated data, we have rewritten this result section, the Material and Methods; we have also introduced a new version of Figure 6 with more details of the quantification process.

- The authors discussed about the orientation of cytonemes. According to the data in Fig. 2 and Fig. 3, orientation of cytonemes appears to be instructed by the location of Ihog expression, and further regulated by the glypican expression. While the cytoneme formation is observed around the clones in Fig. 2, how are the multi-directional cytoneme formation instructed? Does gradient of Ihog or glypicans instruct the direction? Or other components such as chemotactic factors are needed? These are the important issues prior to establishing a model.

Since the overexpression of Ihog stabilizes cytonemes pointing towards regions with lower levels of Ihog, it is not surprising to observe multi-directional cytonemes all around the Ihog overexpressing clones.

The different levels of Ihog and glypicans in the disc epithelium are actually what we argue instruct the cytoneme orientation in the Hh receiving region. Our arguments are based on the hypothesis that punctual levels of Ihog and glypicans in the disc basal surface would be sufficient to guide cytonemes; the experimental and mathematical results validate the model. The results suggest that Ihog, Dally and Dlp are sufficient to guide cytonemes. Nevertheless, this does not imply that these proteins are the only ones affecting orientation. In fact, we are well aware that other proteins could interact with Ihog and glypicans to guide cytonemes. Our future investigation is going in that direction.

We have rewritten the manuscript following the Referees' advice and we have included references suggesting other factors in cytoneme orientation such as those of the EGF and FGF pathways.

Minor points

- S. Fig. 1A: what do "stabilized cytonemes" mean? In the previous paper (2017 eLife), dynamics of cytonemes have been measured. However, I don't find any description of how the authors measure "stability" in either 2017 paper or current manuscript.

The stabilization of cytoneme structures caused by Ihog overexpression was described in González Méndez et al., 2017 and in Bischoff et al., 2013. Cytonemes dynamics "froze" from the wild type lifetime of 10 min to one of up to 9 hours, allowing their visualization. This effect, observed in fixed and non-fixed tissues is what is called stabilization of cytonemes. Since the temporal stabilization of cytonemes has already been published and our work is focused in cytoneme spatial behavior, we did not consider necessary a specific study of cytoneme dynamics. Nevertheless, the *ex vivo*

experiment shown in the new supplementary movie 1 illustrates the temporal cytoneme stabilization in the wing disc. We have rewritten the manuscript following the Reviewer's comments to provide a clearer description of the concept of stabilization used in this work.

- Lane 116: "no stabilized cytonemes are observed", if I understand correctly, the authors only observed fixed tissue image in the wing imaginal disc. The data can be interpreted that generation or stability of cytonemes become less, but time-lapse analysis is needed to conclude which is the case.

We agree with the Reviewer and have tested if the lack of observable cytonemes in those experimental conditions was due to lack of stabilization or the absence of cytonemes; we have performed *in vivo* experiments, which demonstrated that cytonemes were present and dynamic (new Supplementary video 2).

Although the authors have observed cytoneme dynamics in abdominal histoblast, the dynamics of cytonemes in wing imaginal disc may not be the same. Similar arguments are found throughout the paper.

This point has already been discussed above and solved by the *ex vivo* experiment in wing imaginal disc (Supplementary video 1).

- S. Fig. 2B, C: orientation of cytonemes should be statistically analyzed.

We have included a detailed statistical study of lengths (new Figure 2C) and a quantification of the orientation angles of cytonemes (new Figure 2D).

- Fig. 2: Very difficult to understand how the authors analyze the data. According to the picture in Fig. 2A, cytoneme formation might be influenced by the position of the clones (in the center or peripheral of the wing pouch). This has to be discussed.

We have measured the relative distance between clones showing that their distance is important since clones have to be close enough (< 15 microns) to respond to the differences in levels of Ihog and glypicans (cloud analysis and boxplot statistical study in the new Fig 2. B). Our experimental results do not indicate differences in cytoneme behavior depending on the position of the clones in the wing pouch, as also reported by Yang et al. (2021). To avoid a possible misunderstanding and to clarify this point, we have rewritten the results and discussion and included more quantifications (new Figure 2).

- Fig. 6: How is the data processed? There is no clear description in methods. Providing that cytonemes are localized at the basal side of the wing epithelial cells, collecting basal specific Ihog, Dally and Dlp could be useful.

The images in Figure 6 and their quantifications were done in the basal plane. In addition, to ensure that the protein pattern shown is the one exposed to the ECM, extracellular immunostaining was performed as described in Material and Methods.

A more complete information has been included in Figure 6, together with an explanation of how we processed the confocal images to converted Ihog, Dally and Dlp disc surface protein patterns into the numerical matrices used in our mathematical model (see also Material and Methods).

- S. Fig. 1: Details of the data collection are missing. How many samples tested? How were they measured?

We have not included a measurement protocol in this manuscript because it has been published in González-Méndez et al., 2017. We have now included the reference of the origin of these data both in the figure legend and in the manuscript. We have only carried out statistical studies (described in Materials and Methods) related to the stabilized cytoneme phenotype. We have also rewritten this part trying to avoid possible misunderstandings.

- Eq-4 (lane 564) was argued but I could not find Eq-4 in the text.

We have already fixed this mistake.

- Fig. 4 legends are wrong.

The figure legend has been mended.

Reviewer #2 (Remarks to the Author):

This manuscript aims to study the underlying mechanisms of cytoneme behavior by focusing on the Hh signaling cytonemes in *Drosophila* wing image disc. They first showed some experimental evidence by manipulating the levels of Ihog and Glypicans in regions of wing disc cells that the levels of Ihog and glypicans directly affect the cytoneme stabilization, and Ihog and glypicans interact in trans through the Fn1 domain of Ihog to affect cytoneme orientation. Then they developed a mathematical model predicting the effect of Ihog-glypicans interaction in cytoneme orientation.

While the results showing the effect of the levels of Ihog and glypicans on cytoneme behavior is interesting, several critical points remain to be addressed, such as whether Ihog overexpression induced cytoneme formation or stabilized cytonemes? Do those Ihog-induced/stabilized cytonemes from A or P compartment represent Hh signaling cytonemes?

We had previously described that Ihog stabilizes cytoneme but does not influence their formation (González-Méndez et al., 2017). We have also described that there is a spatial and temporal correlation between cytoneme length and Hh gradient formation (Bischoff et al., 2013), what indicates that most of the cytonemes at the A/P compartment border are Hh signaling cytonemes. Nevertheless, we have rewritten some parts of the manuscript to clarify this point.

According to the model proposed here, cytoneme guidance only depends on the levels of Ihog, Dally and Dlp, the question is how the patterns of Ihog, Dally, and Dlp expression levels are created? And how about other factors, such as receptors and ligands? What's their contribution to the cytoneme behavior?

Our experimental results show that Ihog, Dally, and Dlp are involved in cytoneme orientation. Furthermore, mathematical simulations show that these proteins would be sufficient, but that does not mean that cytoneme guidance only depends on these

proteins. We have rewritten the manuscript taking special care to clarify that the mathematical term “sufficient” is not equivalent to “unique”.

It has been described that, in other pathways, ligands and their receptors are usually directly involved in cytoneme behavior (Hatori and Kornberg 2020; Du et al., 2018; Du et al., 2021). So far, we have not observed a direct control of Hh and its receptor Ptc in cytoneme dynamics (González-Méndez et al., 2017). However, the expression patterns of Ihog, Dlp and Dally are regulated in Hh signaling regions, probably due to a modulation by the Hh pathway itself. Therefore, Hh and its receptor might indirectly control cytoneme behavior through modulation of the expression levels of these proteins. We have included a new discussion section addressing this concern.

A recent paper (Yang, Zhang, Yang, et al. eLife 2021;10:e65770) showed interesting results on Ihog mediating cytoneme stabilization through trans-homophilic binding, which is supported by glypicans and occurs through the Fn1 domain of Ihog. Another recent study (Du et al, 2021, BioRxiv) on cytoneme-mediated FGF signaling provides experimental evidence that cytoneme pathfinding is a result of polarized bidirectional signaling response via cytonemes arising from direct FGF-FGFR interactions at the contact sites between the source and recipient cells. Authors should give a better introduction on this and other related findings, and clearly discuss what’s the novelty of their findings on top of the previous studies.

The aforementioned publication (Yang et al, 2021) had already been included in the references and cited several times throughout the work. In the new version of the manuscript, the FGF regulatory loop between signaling and cytoneme extension have been discussed (Du et al., 2018; Du et al., 2021).

Altogether, this paper presented interesting results, predicting a model that might explain one of the ways by which cytonemes are guided for Hh signaling.

Below are some point-to-point comments in detail.

Figure 1 related:

1. Page 4 line 103-108: “In the imaginal wing disc, cytonemes from both Hh producing and receiving cells protrude perpendicularly to the A/P compartment border, as seen following Ihog cis-overexpression (Fig 1A, F and 20105). Thus, Ihog overexpression reveals the spatial cytoneme behavior during the reception process, when cytonemes protruding from the P compartment cells contact with cytonemes from the A compartment cells.” --- Can authors clarify what’s the cytoneme behavior in wing discs without Ihog overexpression?

To address this concern, we have included an experiment expressing different levels of Ihog in the P compartment (overexpression, wild-type, and downregulation) (Supplementary Fig. 2).

Do you see cytonemes orienting toward A/P compartment border in the wt wing disc in live imaging conditions?

Previous works have shown this orientation perpendicular to the A/P compartment border in *in vivo* conditions. We have included the references to address this point.

Ex vivo imaging of live cultured wing disc is feasible. Also, the authors need to explain how the cytoneme behaviors in this study were measured? Are they all done in fixed tissues?

ex vivo imaging in imaginal discs is feasible, although challenging. For this reason, most of the studies of cytoneme dynamics *in vivo* have been performed in abdominal histoblast nests, where it is easier to follow them. However, we agree with the reviewer that it is important to show that a stabilization actually occurs *ex vivo* in the wing disc, and we have included this experiment in the new version of the manuscript (New supplementary Movie 1).

Since this work focuses only on the spatial (non-temporal dynamic) behavior of cytonemes, we have done all the measurements of angles and lengths of the stabilized cytonemes in fixed tissues. We have also included more quantifications in different formats together with statistical studies in the figures.

2. when both confronted A and P cell populations equally express Ihog, there were no cytonemes observed, this result is not equivalent to “no stabilization”.

We agree with the reviewer that under these conditions the non-visualization does not directly imply non-stabilization. To address this concern, we have performed an *in vivo* experiment that confirms that what is observed in fixed tissues is, in fact, non-stabilization (new Supplementary Movie 2).

Figure 2 related:

1. Page 5 line 135-141: “These experiments showed that, besides differential Ihog levels between clones, proximity and size are also important for cytoneme stabilization (Fig. 2B). We again observed that cytonemes protruding from confronting clones both expressing high levels of Ihog could not be stabilized (Fig. 2A), unless the clones were smaller than two cell diameters (Fig. 2C). The length at which a cytoneme can become stable depends on the distance between clones with high levels of Ihog (Fig. 2B).” --- This part of data is not clearly shown by the presented figures. Authors need to present more convincing data for the statements. For example, authors need to show more evidence for each type of interaction, and how many samples and clones were checked? In addition, from the presented image in 2A, it seems that IhogCFP clones polarize cytonemes toward each other from distance, rather than projecting cytonemes in all directions to the region without Ihog expression.

Considering the reviewer's comment about showing more representative cases, we referred in the text to supplementary Figure 2 containing additional cases where the orientation can be observed for all the experimental conditions (high, wild type and low levels of Ihog). We have also developed and included a statistical study of cytoneme lengths and orientation angles in the new Fig. 2. Finally, we have also indicated the number of samples studied in each case.

In the old Figure 2B, each dot on the cloud corresponded to single cytonemes visualized in different clones and in different samples. Since the dotted cloud does not seem to be convincing, we changed representations and included boxplots, statistical studies and quantification of orientation angles.

Lastly, did the authors see any difference in cytoneme behaviors from Ihog-clones in different compartments?

We don't see much differences in behavior between clones induced in the A or in the P compartment. This similar behavior was also independently observed by Yang et al. (2021).

2. Page 5 line 142-146: "Following this line of thought, we could propose that the wild type distribution of Ihog in the wing disc, where it is strongly downregulated in the Hh receiving cells, might suffice to initiate cytoneme orientation from P to A compartment: from regions with high levels of Ihog in the P compartment towards regions with lower levels in the Hh receiving area of the A compartment", --- authors need to provide a reference for this.

For a better comprehension of the Ihog pattern we also included a new Supplementary Movie 3 that shows the pattern from apical to basal and its correlation with the Hh signaling region (marked with Ptc Enhancer RFP). We have also introduced the requested references in the text.

Figure 3 related:

Page 6 line 169-173: "Further suggesting an in trans effect over membrane stability by Ihog overexpression, clones overexpressing glypicans in both P and A compartments showed that P cytonemes are stabilized and oriented when a cell population with high levels of Ihog is in the proximity of another cell population (trans) with high levels of either Dally (Fig. 3C) or Dlp (Fig. 3D)." --- This sentence should be rewritten.

We rewrote the sentence.

Page 7 line 184-200, --- The result, the Ihog Fn1 domain is the site for Ihog-Ihog and Ihog-glypicans trans interactions, has been shown in earlier studies (Yang, Zhang, Yang, et al. eLife 2021;10:e65770). Also, the authors concluded that Ihog-glypicans heterophilic interaction competes with Ihog-Ihog homophilic interaction for cytoneme stabilization, it remains unclear which interaction dominates or has higher affinity. Authors should provide more supporting evidence for this.

Our work suggests that interactions between Ihog, Dally and Dlp may play a role in cytoneme orientation. Experiments expressing a form of Ihog lacking the Fn1 domain indicate that the interaction between these proteins through this domain governs the behavior of cytonemes. The molecular study of the affinities or the biochemistry of these interactions is a topic far from our current research framework. We hope the data obtained can lead to a more thorough study on molecular interactions to regulate guidance.

Lastly, according to Fig. 4A, the number of cytonemes under Ihog-glypicans trans interaction is less than the number of cytonemes in Ihog expression condition like in Fig. 1A, and it is also less than the number of cytonemes with IhogDeltaFn1RFP expression in one side. Why this is the case? can this be explained by the proposed model?

The number of cytonemes seems to change due to the strong Ihog-glypicans interaction that bundles cytonemes; this effect can be observed in clones by looking at the thickness of the cytonemes. To avoid misunderstandings, we have divided the quantification shown in Figure 4 into different plots, so that the reader can be sure that in some cases we are measuring what seems individual cytonemes and in other cases thick cytonemes or bundles of more than one cytoneme. Consistent with this result, we expect Ihog-glypicans heterophilic interaction to dominate over Ihog-Ihog homophilic interaction.

Reviewer #3 (Remarks to the Author):

The authors investigate oriented stabilization as a driving source for the observed directionality of cytonemes, long cellular threads that are required for Hedgehog (Hh) signaling in multiple contexts, including Hh morphogen signaling in the *Drosophila* wing imaginal disc. The paper is largely based on (and guided by) previous data showing that the co-receptor Ihog and the two *Drosophila* HSPGs of the glypican family Dally and Dlp, are required for Hh signaling and stabilize cytonemes. The study adds on the spatial requirements for these factors and shows that differences in the relative levels of Ihog and Dally/Dlp are instructive for the stabilization and orientation of cytonemes. Specifically, they find that cytonemes are stabilized, and potentially oriented, when cells of the epithelium display differences in the Ihog levels and experimentally flattening such differences cause loss (or destabilization) of cytonemes. Glypicans, previously shown to interact with and affect surface stability of Ihog, do also affect cytoneme stability and orientation in trans. This effect is mediated by their interactions with Ihog, suggesting that the relative levels of the players are important for cytonemes stabilization and orientation. The authors back up their data with an elegant mathematical model (to my knowledge the first application of mathematical modeling in the cytonemes field) which supports the findings and makes the prediction, that the physiologically occurring relative levels distribution of the components (Ihog, Dally, Dlp) could suffice to generate the observed cytoneme orientation and alignment.

Major points.

Many of the data shown confirm or expand already previous findings (in part from the same group). The *in vivo* studies are very elegant and the tools used superb. In all experiments, the eventual read-out is the presence (i.e. stabilization of cytonemes) using fluorescent Ihog as a proxy. Thus, in all cases, overexpressed Ihog-GFP (or RFP/CFP) serves a dual function: It stabilizes cytonemes so that they become (easily) detectable, and at the same time it is used as a read-out for these structures. In other words, we are not looking at “wild-type” cytonemes, but rather at cytonemes induced by non-physiologically high levels of Ihog. While I understand that cytonemes of wild-type cells (in this case cells that are not overexpressing factors that impact on cytoneme dynamics) are hard to observe, I believe it would be essential to close the circle and test in at least in one assay the predictions of the models by monitoring cytonemes emanating from cells that are not additionally modified in their capacity to produce or stabilize cytonemes.

Experimentally, we show that cytonemes labelled with markers other than Ihog are also oriented perpendicular to the A/P compartment border (new supplementary figure 2 with cytonemes labeled with CD4). Following the reviewer suggestion, to close the circle, we included wild type experimental orientation in the new figure 7 and compare it with the model predictions. Also a more detailed study of the temporal wild type behavior can be observed in movie 3. We have rewritten the manuscript to clarify this issue.

Similarly, I miss some direct connection to graded Hh signaling itself using markers of the pathway.

We have included a new Supplementary movie 3 that analyses the correlation of Ihog pattern with the graded expression of the Hh receptor Patched (Ptc). In the source data excel file we have also incorporated the quantified Ihog, Dlp and Dally patterns together with the spatial correlation with Hh signal responses Ptc and Ci for more than 17 imaginal wing discs. However, the complete study of this connection of cytoneme dynamics with the graded Hh signaling has been the subject of an *in silico* model for Hh gradient formation mediated by cytonemes recently published (Aguirre-Tamaral & Guerrero, 2021). We have introduced this reference in the new version manuscript.

I am not sure if I understand the interpretation of the experiments with IhogDeltaF1: Is this construct biologically inert?

Our guiding model is based on the assumption that Ihog, Dally, and Dlp interaction is mediated by the Fn1 domain for cytoneme orientation. The Ihog Δ Fn1 mutant protein, lacking the FN1 domain, was used in Simon et al., 2021, showing that the overexpression of this Ihog mutant form does not stabilize cytoneme but remains functional for other Ihog roles.

Can this construct when used in I the simplest form of the assays used here (hh>IhogDeltaFn1 similar to 1A) stabilize cytonemes ? If not, then the interpretation of 4C is trivial (or just confirmatory that the protein is not functional).

We and others have recently reported that cytonemes protruding from cells expressing this construct are not stabilized (Yang et al. 2021 and Simon et al. 2021) because its interaction with glypican is abolished. However, this construct still interacts with other proteins of the Hh pathway (Simon et al. 2021) and Shifted and Ptc (unpublished results). Additionally, the objective of the experiment was to show the non-autonomous effect produced on the cytonemes protruding from cells in the compartment opposite to the one in which the construct is expressed. This result indicates that the trans-interaction governs the stabilization of the cytonemes.

Other/Minor points

Line 29 abstract: "A paradigmatic example" sounds redundant (= "An exemplary example"), consider "A prominent/classic/iconic example or paradigm"

Yes, we agree with the reviewer that the sentence is redundant; we have corrected it.

Figure 1 B and F: Here they use *ptcGal4* instead of the *dpp* driver used throughout the study. Is there a reason for that? I guess both drivers express in the same cells. Neither of the drivers appears in the material section.

Dpp-Lex and *Ptc-Gal4* are drivers used to induce expression specifically in the receptor cells of the anterior compartment of the wing disc. We use one or the other for a specific experiment depending on the genetic tools available.

We thank the reviewer for noticing these missing references in the Material and Methods section, we have included them in the new version.

Line 167: “navigating within glypican-def clones” this is unclear to me, do they mean cytonemes stemming from glypican-deficient clones”?

We have rewritten the sentence to make clear that cytonemes stem from glypican-deficient clones.

Sentence starting at line 195: Fix syntax

We have rewritten this sentence.

REVIEWER COMMENTS

Reviewer #1 (Remarks to the Author):

The manuscript has been significantly improved. Employing an ex vivo experiment is useful to fill out the gap between abdominal histoblast and wing imaginal disc. The orientation of cytonemes is reasonably addressed. Overall, the revised paper addressed most of my questions and provides the logical model. I support the publication of the paper.

Reviewer #2 (Remarks to the Author):

The revised paper has addressed most of the questions/suggestions. Additional supporting results are presented. Results clearly suggest a role of iHog and glypican interactions on the stability or production of cytonemes, and the work has proposed a mathematical model of cytoneme guidance based on the iHog-glypican interactions.

I have a few additional suggestions for the new version, especially on the text describing some experimental results.

1) A major concern is on the presentation of the Figure 2 results, in particular that described the levels and location of iHog production deciding the directionality of cytonemes. Unlike images in Fig 1A, where we can see only one side of the expressing cells, the clonal analyses revealed more. Clearly, an iHog overexpressing clone can emanate/stabilize cytonemes in all directions (A/P/D/V) without any apparent bias (e.g., blue arrow clones). So, iHog can stabilize cytonemes without any directionality/orientational bias. However, when one iHog overexpressing clone touches another, apparently, there is an effect on the stability or production of cytonemes at the interface. These results are interesting from the perspective of the probable homotypic iHog-iHog interactions that might control cytoneme stability/production. However, these results should not suggest its role in initiating/deciding cytoneme orientation/polarity. Therefore, a suggestion is that the paper should focus on the cytoneme stability that the iHog expression provides. A predictive theoretical model on cytoneme guidance based iHog-iHog or iHog-glypican is understandable, but the experimental data suggests only an effect on cytoneme stability.

Several parts in the main text/Fig legends suggest an effect on cytoneme polarity, change in polarity, or initiation of polarity during cytoneme growth, which might not be correct. For instance:

a. Fig 2 legend title, "Cytoneme stabilization and orientation depend on the distance to Ihog source", authors might be suggesting the probability of cytoneme extension/production instead of cytoneme orientation.

b. Figure 2D. Legend: "We studied in detail cytonemes over the green dotted line (i.e., cytonemes longer than the distance with the closest clone), and we observed that those cytonemes were able to grow since they change their orientation () to avoid the confronted clones (= 0 means no change in orientation and = 90 opposite direction). Scale bars: 15µm."

c, Line 133: "Cytoneme length is affected by the proximity of another clone if it is close enough (less than 15 microns Fig. 2C). This effect becomes statistically stronger the shorter the distance (Fig. 2C); actually, cytonemes in close proximity to a clone can only grow if they change their orientation to circumvent the confronted clone (Fig 2.D)."

c. Line 150: "initiate" cytoneme orientation.

I think that the cytoneme guidance/selective polarization is not well supported by the experiments shown. Authors have studied fixed steady-state tissues. So, not sure how we can predict that the

growing cytonemes change directions to avoid a confronted iHog clone. Based on Fig 2A, iHog clones emanate cytonemes from all open sides, indicating the role of iHog in cytoneme stability but not in selective polarity. These images suggested that cytoneme stability/production is affected at the clone-clone interface within a certain range of the inter-clonal distance. Other sides of the clones facing the native iHog-expressing regions remained unaffected. This might indicate an important role of iHog-iHog trans-interactions on cytoneme stability/production at the cell-cell interface, but its role in cytoneme orientation cannot be assessed from these data.

The quantitative analyses in Fig 2B,C, D suggest an effect on cytoneme stability, but these are dependent on the interpretation of the images like Fig. 2A. From the perspective of cytoneme stability, it might be important to also quantitate the number of cytonemes to indicate any changes in the stability at the interface, in addition to the length of the existing cytonemes. A change in cytoneme length is expected to be relative to the distance of the target. These analyses do not suggest an effect on the orientation/polarity of cytonemes. In Fig.2 A, some clones show partially circumventing each other, but the presented images are not sufficient to suggest a change in the direction of growing cytonemes, unlike what is written. For the inter-clone crossing expts, there are no data on the control WT clones to assess a relative change in the crossing frequency. 18% of the iHog clones can cross each other via thick membrane projections/cytonemes, and the process could be affected by the clone position within the disc. Even if there is a relative reduction of clones crossing each other in comparison to the control clones, it won't be easy to predict that the growing iHog-cytonemes can change directions to circumvent the confronted clone without live imaging analyses. While additional experiments are not required, conclusions on cytoneme orientation need to be toned down.

2) Another concern is about the levels of iHog. Line-130 (expt) and Line 137-147 (conclusion): "Altogether, these results indicate that differences in Ihog levels between cytoneme membranes in trans play a decisive role in cytoneme dynamics." This statement gives the impression that different levels of iHog control behaviors. However, neither the RNAi-mediated knockdown nor the over-expression by different drivers are well-controlled changes in the expression levels. Moreover, membrane-surface levels of these CAMs, which might be required for their activity, were not verified. A comparison of the inter-cytoneme iHog-iHog interactions between clones under control and test genotypes might be required to convincingly suggest a decisive role of the cell surface levels of iHog on cytonemes. Secondly, genetic perturbations of an important molecule like iHog can have pluripotent effects. So, the conclusive sentences based on these experiments should be more cautious.

Minor points:

1) Line 80:

"The dynamics of Hh cytonemes over time has already been extensively studied 11,15,19 together with the correlation of cytoneme dynamics with the morphogen signaling gradient 12,20."

To my understanding, the dynamic behavior of very thin Hh-cytonemes is very challenging to capture within a complex tissue. So, Hh cytoneme dynamics were best studied under iHog-overexpressing conditions, which might or might not be similar to the WT and might also be variable in different tissues if they have different levels/types of glypicans/HSPGs and iHog. So, considering this, the word "extensively studied" might not be appropriate.

2) "that Ihog overexpression in the same cells (overexpression in cis) stabilizes the dynamics of approximately 70% of the cytoneme population."

This sentence might need rewriting. Ihog overexpression stabilizes cytonemes might be better.

3) Line 124: "To ascertain that this apparent lack of cytonemes is due to non-stabilization instead of absence, we performed in vivo experiments in abdominal histoblast nets (Supplementary Movie 2)."

Some typos can be corrected, and the above paragraph ends abruptly. The new result on histoblast nest indeed suggested that the lack of cytoneme detection in fixed tissue is due to unstable/dynamic cytonemes. The experiment and results on histoblast should be briefly described in the main text. Also, it is unclear if the responses in the disc and histoblast nest are equivalent. In Suppl. movie 2, the A compartment of histoblast apparently showed more cytonemes than P. Secondly, the dynamic cytoneme projection-retraction cycles under the iHog overexpression condition seem to oppose the fact that iHog overexpression stabilizes histoblast cytonemes.

4) No. of discs and/or clones can be indicated in the Figure legends.

5) Line 140 and related Suppl. Fig indicating "Lifeactin" and it is a "neutral" label. The term neutral is confusing; not sure if it indicates the physical nature of the label. Secondly, I am unfamiliar with the "Lifeactin" label. Does the text refer to the actin-binding peptide, Lifeact, (not Lifeactin)?

6) Line 174: " cytonemes are stabilized in bundles and oriented when a cell population with high levels of Ihog is in the proximity of another cell population (trans) with high levels of either Dally (Fig. 3C) or Dlp (Fig. 3D)".

A role in determining the orientation of cytonemes is not conclusive based on the experiment shown. It is important to focus on the control of the stability of cytonemes that were already polarized. A probable effect on orientation/guidance can be discussed and modeled for future validation. The knowledge of molecular interactions determining cytonemes polarity/guidance is in its infancy.

Reviewer #3 (Remarks to the Author):

The revised version of the manuscript incorporates new data and adequately answers most point of concern. In my opinion, the manuscript describe an innovative, integrative approach to study cytonemes dynamics, provides new insights in the field and has improved through the revision process. I have only some, in part very minor, points that may need fixing and/or clarification :

1. Please better describe Supplementary Figure 1, I understand that the data comes from quantification of already published data (histoblast nests). Also, please better detail the tools used here: GMA might not be clear to non-specialists, even if looked up in the Material & Methods sections

2. Line 124, I think this sentence should be followed by the result of the experiment for better understanding

3. Panels D and E are not discussed and do not appear in the main body text . Same for panels D and E of Fig. 4

4. Line 197, "in contrast to the observed"

5. I am not sure if I completely understand the calculation of the surface distribution of proteins in Fig. & panel E (my apologies that I did not picked this up in my first review): How comes that the characteristic stripe of strong Dally expression along the D-V compartment boundary does not show up in the graphic representation? Is this region omitted from the analysis? I have the impression that the decrease of Dlp levels in the same area is captured in the "3-D" surface representation.

REVIEWER COMMENTS

Reviewer #1 (Remarks to the Author):

The manuscript has been significantly improved. Employing an *ex vivo* experiment is useful to fill out the gap between abdominal histoblast and wing imaginal disc. The orientation of cytonemes is reasonably addressed. Overall, the revised paper addressed most of my questions and provides the logical model. I support the publication of the paper.

We want to thank the referee for his/her the positive feedback in the previous version that have helped us to improve our manuscript.

Reviewer #2 (Remarks to the Author):

The revised paper has addressed most of the questions/suggestions. Additional supporting results are presented. Results clearly suggest a role of iHog and glypican interactions on the stability or production of cytonemes, and the work has proposed a mathematical model of cytoneme guidance based on the iHog-glypican interactions.

I have a few additional suggestions for the new version, especially on the text describing some experimental results.

We want to thank the referee for his/her positive feedback, comments and corrections that have helped us to improve our manuscript.

The main argument of the Referee 2 is mostly related to our interpretation of the results shown in Fig2. We thank the reviewer for his sharp analysis of the results shown in this figure. We agree with his/her interpretation, which does not invalidate the conclusions of our work, and have rewritten some sections of the manuscript accordingly.

1) A major concern is on the presentation of the Figure 2 results, in particular that described the levels and location of iHog production deciding the directionality of cytonemes. Unlike images in Fig 1A, where we can see only one side of the expressing cells, the clonal analyses revealed more. Clearly, an iHog overexpressing clone can emanate/stabilize cytonemes in all directions (A/P/D/V) without any apparent bias (e.g., blue arrow clones). So, iHog can stabilize cytonemes without any directionality/orientational bias. However, when one iHog overexpressing clone touches another, apparently, there is an effect on the stability or production of cytonemes at the interface. These results are interesting from the perspective of the probable homotypic iHog-iHog interactions that might control cytoneme stability/production.

As we mentioned in the manuscript, lhog overexpressing clones can stabilize cytonemes in all directions if they are far enough from each other. However, we have also observed that the stabilization depends on the distance between lhog expressing cells and cytonemes cannot be stabilized if the lhog overexpressing clones are close, with the exception of the cytonemes that stabilize surrounding clones. Quantifications shown in figure 2 describe the effect of lhog levels in clones that are touching each other and

clones that are located up to 15 microns of distance (around 3-4 cells from the Ihog source). These effects have been quantified and shown to be statistically significant (figure 2 C and D).

We agree with the referee that the results shown in this Fig 2 about cytoneme behavior at the interface of touching clones indicate that the levels of the homotypic Ihog-Ihog interactions may control cytoneme stability.

However, these results should not suggest its role in initiating/deciding cytoneme orientation/polarity. Therefore, a suggestion is that the paper should focus on the cytoneme stability that the iHog expression provides. A predictive theoretical model on cytoneme guidance based iHog-iHog or iHog-glypican is understandable, but the experimental data suggests only an effect on cytoneme stability.

The results presented in figure 2, figure 3, figure 4 and supplementary figures 3 and 4, showed alteration of cytoneme orientation (quantified in figure 2). The presented mathematical simulations also supported the Ihog and glypicans effect in orientation (as showed in figures 5 and 7 and videos 1, 2 and 3). Therefore, we disagree with the reviewer in that our results do not suggest a role for Ihog and glypicans interaction in cytoneme orientation.

Several parts in the main text/Fig legends suggest an effect on cytoneme polarity, change in polarity, or initiation of polarity during cytoneme growth, which might not be correct. For instance:

a. Fig 2 legend title, "Cytoneme stabilization and orientation depend on the distance to Ihog source", authors might be suggesting the probability of cytoneme extension/production instead of cytoneme orientation.

Although Fig 2 showed cytoneme stabilization measurements and also quantifications of their orientation (fig 2D), we agree with the referee and we have changed the Fig.2 title to "Cytoneme behavior depends on the distance to Ihog source", and we have also changed in the text and in the legend of Fig.2 the possible misunderstanding of the role of Ihog by itself in cytoneme polarization/orientation.

b. Figure 2D. Legend: "We studied in detail cytonemes over the green dotted line (i.e., cytonemes longer than the distance with the closest clone), and we observed that those cytonemes were able to grow since they change their orientation () to avoid the confronted clones (= 0 means no change in orientation and = 90 opposite direction). Scale bars: 15 μ m."

We have modulated the sentence to ...and we observed that those cytonemes that are able to stabilize show a change in their orientation...

c, Line 133: "Cytoneme length is affected by the proximity of another clone if it is close enough (less than 15 microns Fig. 2C). This effect becomes statistically stronger the shorter the distance (Fig. 2C); actually, cytonemes in close proximity to a clone can only grow if they change their orientation to circumvent the confronted clone (Fig 2.D)."

We have also modulated this sentence: ...actually, cytonemes in close proximity to a clone can only be stabilized if they have an orientation that circumvents the confronted clone (Fig 2.D)."

c. Line 150: "initiate" cytoneme orientation.

We agree with the referee. We have changed this sentence.

I think that the cytoneme guidance/selective polarization is not well supported by the experiments shown. Authors have studied fixed steady-state tissues. So, not sure how we can predict that the growing cytonemes change directions to avoid a confronted iHog clone.

We agree with the referee that we cannot predict the orientation of cytonemes only based on the data shown in Fig 2 and supplementary Fig 3. However, these results suggest that confronted high levels of Ihog cannot stabilize cytonemes and the cytonemes that are stabilized are those that have a trajectory circumventing the clone. Similar results are shown in Fig 1. We have rewritten the text to make these points clear.

Based on Fig 2A, iHog clones emanate cytonemes from all open sides, indicating the role of iHog in cytoneme stability but not in selective polarity.

We coincide with this assertion: since Ihog-Ihog interaction in trans is important for cytoneme stabilization, Ihog clones emanate cytonemes from all open sides but only when facing a wild type territory with lower levels of Ihog. These results also suggest that other molecular interactions together with Ihog are needed to orient cytonemes. We have rewritten the results section to make this point clearer.

These images suggested that cytoneme stability/production is affected at the clone-clone interface within a certain range of the inter-clonal distance. Other sides of the clones facing the native iHog-expressing regions remained unaffected. This might indicate an important role of iHog-iHog trans-interactions on cytoneme stability/production at the cell-cell interface, but its role in cytoneme orientation cannot be assessed from these data.

We have rewritten the text accordingly to referee comments.

The quantitative analyses in Fig 2B,C, D suggest an effect on cytoneme stability, but these are dependent on the interpretation of the images like Fig. 2A.

Based on the confocal images in Fig 2 and 4 and Supplementary Fig 3, the interpretation could be either that cytonemes with high levels of Ihog change orientation to circumvent the ectopic Ihog clones or that only those cytonemes that surround the Ihog clones are the ones that stabilize. Therefore, since we can ensure which of the two interpretations is the real one for Fig. 2A, we have rewritten the text accordingly.

From the perspective of cytoneme stability, it might be important to also quantitate the number of cytonemes to indicate any changes in the stability at the interface, in addition to the length of the existing cytonemes.

We have already quantified in Figure 4 the number of cytonemes at the interface, together with a statistical study versus a wildtype interface (control).

A change in cytoneme length is expected to be relative to the distance of the target. These analyses do not suggest an effect on the orientation/polarity of cytonemes.

We agree with the reviewer that the changes in the length Fig 2B and C only demonstrate that Ihog-Ihog interactions affect the length of stabilized cytonemes. For that reason, we also included analysis of the orientation angle in Fig 2.D.

In Fig.2 A, some clones show partially circumventing each other, but the presented images are not sufficient to suggest a change in the direction of growing cytonemes, unlike what is written. For the inter-clone crossing expts, there are no data on the control WT clones to assess a relative change in the crossing frequency.

The relative change in the crossing frequency in wild type has already been shown in the confocal experiments of the supplementary Figure 3.

18% of the iHog clones can cross each other via thick membrane projections/cytonemes, and the process could be affected by the clone position within the disc. Even if there is a relative reduction of clones crossing each other in comparison to the control clones, it won't be easy to predict that the growing iHog-cytonemes can change directions to circumvent the confronted clone without live imaging analyses. While additional experiments are not required, conclusions on cytoneme orientation need to be toned down.

In several places along the manuscript we clearly state that Ihog-Ihog homophilic interaction in different cell populations is not sufficient for cytoneme orientation. We therefore think that we have toned down our conclusions on the role of Ihog on cytoneme stability versus orientation.

2) Another concern is about the levels of iHog. Line-130 (expt) and Line 137-147 (conclusion): "Altogether, these results indicate that differences in Ihog levels between cytoneme membranes in trans play a decisive role in cytoneme dynamics." This statement gives the impression that different levels of iHog control behaviors.

Experiments inducing clones with different levels of Ihog support this assertion.

However, neither the RNAi-mediated knockdown nor the over-expression by different drivers are well-controlled changes in the expression levels. Moreover, membrane-surface levels of these CAMs, which might be required for their activity, were not verified. A comparison of the inter-cytoneme iHog-iHog interactions between clones under control and test genotypes might be required to convincingly suggest a decisive role of the cell surface levels of iHog on cytonemes.

The experiments in supplementary Fig. 3 already showed the behavior of cytonemes in different genotypes requested by the reviewer.

Secondly, genetic perturbations of an important molecule like iHog can have pluripotent effects. So, the conclusive sentences based on these experiments should be more cautious.

Aware that Ihog can have pluripotent effects, we rewrote part of the conclusions trying to be more cautious.

Minor points:

1) Line 80:

"The dynamics of Hh cytonemes over time has already been extensively studied 11,15,19 together with the correlation of cytoneme dynamics with the morphogen signaling gradient 12,20."

To my understanding, the dynamic behavior of very thin Hh-cytonemes is very challenging to capture within a complex tissue. So, Hh cytoneme dynamics were best studied under iHog-overexpressing conditions, which might or might not be similar to the WT and might also be variable in different tissues if they have different levels/types of glypicans/HSPGs and iHog. So, considering this, the word "extensively studied" might not be appropriate.

We have changed this sentence to make it more accurate.

2) "that Ihog overexpression in the same cells (overexpression in cis) stabilizes the dynamics of approximately 70% of the cytoneme population."

This sentence might need rewriting. Ihog overexpression stabilizes cytonemes might be better.

We have corrected the original sentence to ...the stabilization of cytonemes by overexpression of Ihog is around 70% of the total cytoneme population (González-Méndez et al., 2017)...

3) Line 124: "To ascertain that this apparent lack of cytonemes is due to non-stabilization instead of absence, we performed in vivo experiments in abdominal histoblast nets (Supplementary Movie 2)."

Some typos can be corrected, and the above paragraph ends abruptly.

We have corrected these mistakes in the new revised version.

The new result on histoblast nest indeed suggested that the lack of cytoneme detection in fixed tissue is due to unstable/dynamic cytonemes. The experiment and results on histoblast should be briefly described in the main text. Also, it is unclear if the responses in the disc and histoblast nest are equivalent. In Suppl. movie 2, the A compartment of histoblast apparently showed more cytonemes than P. Secondly, the dynamic cytoneme projection-retraction cycles under the iHog overexpression condition seem to oppose the fact that iHog overexpression stabilizes histoblast cytonemes.

To induce expression in both compartments in the same sample, we have to use two different drivers (Gal4 and LexA). Nevertheless, the two type of drivers do not induce the

same amount of protein and the differences observed are due to their expression levels. Previous works (González-Méndez et al., 2017; 2020) support this assertion.

4) No. of discs and/or clones can be indicated in the Figure legends.

We have included this information in the new revised version.

5) Line 140 and related Suppl. Fig indicating "Lifeactin" and it is a "neutral" label. The term neutral is confusing; not sure if it indicates the physical nature of the label. Secondly, I am unfamiliar with the "Lifeactin" label. Does the text refer to the actin-binding peptide, Lifeact, (not Lifeactin)?

Yes, it was our mistake and we have corrected the notation to lifeact for the actin-binding peptide. We have used lifeact to label the wild type cytoneme behavior because its expression has been reported not to affect filopodial behavior (Riedl, et al. (2010), new reference 23 in the manuscript). We have rewritten this sentence to make it more clear.

6) Line 174: " cytonemes are stabilized in bundles and oriented when a cell population with high levels of Ihog is in the proximity of another cell population (trans) with high levels of either Dally (Fig. 3C) or Dlp (Fig. 3D)".

A role in determining the orientation of cytonemes is not conclusive based on the experiment shown. It is important to focus on the control of the stability of cytonemes that were already polarized.

The experiments confronting clones expressing Dally or Dlp with those expressing Ihog, show cytonemes stabilized and oriented towards each other. These results suggest that the interaction of Ihog and glypicans in trans could have a role in cytoneme orientation in addition to its role in cytoneme stabilization. Orientation could occur before or at the same time than stabilization. Possibly, changes in the adhesion could create regions with higher cytoneme stability and this could drive their polarization.

A probable effect on orientation/guidance can be discussed and modeled for future validation.

Experimental data shown in figures 5 and 7, and videos 1 and 2 compare the model predictions with experimental cases. Indeed, a specific validation of the guidance model for real cases in overexpression conditions is shown in figure 5.

The knowledge of molecular interactions determining cytonemes polarity/guidance is in its infancy.

We also agree that the knowledge of molecular interactions determining cytonemes polarity/guidance is in its infancy. Our results propose that the interaction of these proteins could play a role in the orientation, opening future research lines.

Reviewer #3 (Remarks to the Author):

The revised version of the manuscript incorporates new data and adequately answers most point of concern. In my opinion, the manuscript describe an innovative, integrative approach to study cytonemes dynamics, provides new insights in the field and has improved through the revision process. I have only some, in part very minor, points that may need fising and/or clarification:

We want to thank the referee for his/her positive feedback, comments and corrections that have helped us to hopefully improve the manuscript.

1. Please better describe Supplementary Figure 1, I understand that the data comes from quantification of already published data (histoblast nests). Also, please better detail the tools used here: GMA might not be clear to non-specialists, even if looked up in the Material & Methods sections

To facilitate the comprehension of the results, we have introduced a better description of the genetic tool used.

2. Line 124, I think this sentence should be followed by the result of the experiment for better understanding

Following this suggestion, we have rewritten this sentence.

3. Panels D and E are not discussed and do not appear in the main body text . Same for panels D and E of Fig. 4

It was our mistake; we have now described those panels in the main text.

4. Line 197, “in contrast to the observed”

We have rewritten the sentence.

5. I am not sure if I completely understand the calculation of the surface distribution of proteins in Fig.6 & panel E (my apologies that I did not picked this up in my first review): How comes that the characteristic stripe of strong Dally expression along the D-V compartment boundary does not show up in the graphic representation? Is this region omitted from the analysis? I have the impression that the decrease of Dlp levels in the same area is captured in the “3-D” surface representation.

Since we are interested in cytoneme orientation in the Hh signaling area at the A/P compartment border, the angle in the 3D view in the graphic representation was selected to better show the protein distribution pattern at the A/P compartment area. The regions marked in blue in the experimental panels A, B and C correspond to the ROI areas selected to analyze the statistics based on the 1D plots. Nevertheless, the data of the entire wing pouch were used to simulate cytoneme orientation in the wild type wing disc (Figure 7 and video 3). Indeed, our simulations suggest that the strong Dally regulation in the D-V boundary region could be responsible for cytoneme orientation of those signaling pathways acting in the orientation perpendicular to D-V compartment boundary.

REVIEWERS' COMMENTS

Reviewer #2 (Remarks to the Author):

The authors have addressed my concerns in the revised manuscript.

Reviewer #3 (Remarks to the Author):

The revised manuscript adequately addressed open points ! I am in support of publishing this study in its present form

Final REVIEWER COMMENTS

REVIEWERS' COMMENTS

Reviewer #2 (Remarks to the Author):

The authors have addressed my concerns in the revised manuscript.

We want to thank the referee for his/her the positive feedback in the previous version that have helped us to improve our manuscript.

Reviewer #3 (Remarks to the Author):

The revised manuscript adequately addressed open points ! I am in support of publishing this study in its present form.

We want to thank the referee for his/her the positive feedback in the previous version that have helped us to improve our manuscript.